# A mechanistic model of an upper bound on oceanic carbon export as a function of mixed layer depth and temperature

Zuchuan Li[*], Nicolas Cassar

Division of Earth and Ocean Sciences, Nicholas School of the Environment, Duke University, Durham, North Carolina, USA

[*] Corresponding author: Zuchuan Li (zuchuan.li@duke.edu)

**Key points**

1. A mechanistic model of an upper bound on carbon export is developed based on the metabolic balance of photosynthesis and respiration in the oceanic mixed layer

2. Using parameters available in the literature, the modeled upper bound envelopes field observations of export production estimated from $^{234}$Th and sediment traps and $O_2$/Ar-derived net community production

3. The model identifies regions of the Southern Ocean where carbon export is likely limited by light during part of the growing season

**Abstract**

Export production reflects the amount of organic matter transferred from the surface ocean to depth through biological processes. This export is in great part controlled by nutrient and light availability, which are conditioned by mixed layer depth (MLD). In this study, building on Sverdrup's critical depth hypothesis, we derive a mechanistic model of an upper bound on carbon export based on the metabolic balance between photosynthesis and respiration as a function of MLD and temperature. We find that the upper bound is a positively skewed bell-shaped function of MLD. Specifically, the upper bound increases with deepening mixed layers down to a critical depth, beyond which a long tail of decreasing carbon export is associated with increasing heterotrophic activity and decreasing light availability. We also show that in cold regions the upper bound on carbon export decreases with increasing temperature when mixed layers are deep, but increases with temperature when mixed layers are shallow. A metaanalysis shows that our model envelopes field estimates of carbon export from the mixed layer. When compared to satellite export production estimates, our model indicates that export production in some regions of the Southern Ocean, most particularly the Subantarctic Zone, is likely limited by light for a significant portion of the growing season.

**Key words**: Export production, net community production, upper bound, mixed layer depth, temperature

## 1. Introduction

Photosynthesis in excess of respiration at the ocean surface leads to the production of organic matter, part of which is transported to the deep ocean through sinking and mixing (Volk and Hoffert, 1985). This biological process, known as export production (aka soft tissue biological carbon pump) lowers carbon dioxide ($CO_2$) concentrations at the ocean surface and facilitates the flux of $CO_2$ from the atmosphere into the ocean (Falkowski et al., 1998; Ito and Follows, 2005; Sigman and Boyle, 2000).

Export production is frequently assumed to be a function of net community production (NCP) which is defined as the balance between net primary production (NPP) and heterotrophic respiration (HR), or the difference between gross primary production (GPP) and community respiration (CR; HR plus autotrophic respiration (AR)) (the acronyms used in this study are presented in Table 1) (Li and Cassar, 2016):

$$CO_2 + H_2O \xrightleftharpoons[\underbrace{\xrightarrow{NCP} \xleftarrow{HR}}_{NPP}]{GPP} \xleftarrow{AR} Organic\ matter + O_2 \qquad (1)$$

$$Export\ production = NCP - MLD \times \frac{d(POC + DOC)}{dt} \qquad (2)$$

where POC, DOC and MLD represent particulate organic carbon, dissolved organic carbon and mixed layer depth, respectively. If the organic carbon inventory (POC+DOC) in the mixed layer is at steady state, NCP is equal to export production (equation (2)). Without allochthonous sources of organic matter, if the organic matter inventory in the mixed layer decreases, NCP will be predicted to be transiently smaller than export production. Conversely, export may lag NPP (Henson et al., 2015; Stange et al., 2017), in which case NCP is expected to be greater than export production.

Net community production is in great part regulated by the availability of nutrients and light.
Light availability exponentially decays with depth due to absorption by water and its constituents.
The mixing of phytoplankton to depth therefore impacts phytoplankton physiology and
productivity (Cullen and Lewis, 1988; Lewis et al., 1984), with the depth-integrated NPP expected
to increase down to the euphotic depth. Respiration, on the other hand, is often modeled to be some
function of organic matter concentration, which is expected to be constant with depth if
homogenously mixed within the mixed layer. Temperature is also believed to be an important
control on carbon export because respiration is more temperature-sensitive than photosynthesis
(Laws et al., 2000; López-Urrutia et al., 2006; Rivkin and Legendre, 2001). Field observations
confirm that NCP is generally lower at high temperatures and consistently low when mixed layers
are deep. These patterns have been attributed to the balance between depth-integrated
photosynthesis (controlled by the availability of nutrients and light) and respiration as a function
of MLD and temperature (Cassar et al., 2011; Eveleth et al., 2016; Huang et al., 2012; Shadwick
et al., 2015; Tortell et al., 2015). However, descriptions of the underlying mechanisms heretofore
remain qualitative. Likewise, the effects of light and nutrient on carbon fluxes are difficult to
disentangle. For example, high-nutrient, low-chlorophyll regimes in the Southern Ocean have been
attributed to iron limitation (Boyd et al., 2000), deep mixed layers and light limitation (Nelson and
Smith, 1991; Mitchell and Holm-Hanse, 1991; Mitchell et al., 1991), or both (Sunda and Huntsman,
1997). To decompose the influence of light and nutrient availability on NCP, we define the upper
bound on carbon export from the mixed layer ($NCP^*$) as the maximum export achievable should
all limiting factors other than light (taking into account self-shading) be alleviated.
In his seminal paper, Sverdrup presented an elegant model to demonstrate that vernal
phytoplankton blooms (i.e., organic matter accumulation at the ocean surface) may be driven by
increased light availability when the MLD shoals above a critical depth ($Z_c$) (Sverdrup, 1953). In
our study, we build upon Sverdrup (1953) and derive a mechanistic model of an upper bound on
carbon export based on the metabolic balance of photosynthesis and respiration in the oceanic
mixed layer, where the metabolic balance is derived from MLD, temperature, photosynthetically
active radiation (PAR), phytoplankton maximum growth rate ($\mu_{max}$), and heterotrophic activity.
Our approach is analogous to other efforts where mechanistic models were derived to predict
proxies of carbon export (e.g., Dunne et al. (2005) and Cael and Follows (2016)). We compare our
$NCP^*$ model to observations, and use this model in conjunction with satellite export production
estimates to identify regions in the world's oceans where light may limit export production. Our
key findings are that 1) using parameters available in the literature, the modeled upper bound
envelopes field observations of export production estimated from [234]Th and sediment traps and
$O_2$/Ar-derived NCP, and 2) the model identifies regions of the Southern Ocean where carbon
export is likely limited by light during part of the growing season.
**2. Model description and comparison to observations**
**2.1. Net community production and light availability**
A conceptual representation of the metabolic balance between volumetric NCP, NPP, and HR
profiles is presented in Figure 1(A). According to equation (1), the volumetric NCP flux at a given
depth ($z$) in the mixed layer results from the difference between volumetric NPP and HR:
$$NCP(z) = NPP(z) - HR(z) \quad (3)$$
where $z$ increases with depth. $NPP(z)$ is a function of the autotroph's intrinsic growth rate
($\mu$) times their biomass concentration ($C$). Assuming that the effect of nutrients and light on
photosynthetic rates abides by Michaelis-Menten kinetics, and neglecting the effect of
photoinhibition (Dutkiewicz et al., 2001; Huisman and Weissing, 1994), $NPP(z)$ may be
expressed as follows:

$$NPP(z) = \mu(z) \times C = \frac{N}{N + k_m^N} \times \frac{I(z)}{I(z) + k_m^I} \times \mu_{max} \times C \qquad (4)$$

where $\mu_{max}$ is the maximum intrinsic growth rate of the autotrophic community; $N$ and $k_m^N$
represent the nutrient concentration and half-saturation constant, respectively; and $I$ and $k_m^I$
represent the irradiance level and half-saturation constant, respectively. $\mu_{max}$, $N$, $k_m^N$, $k_m^I$ and $C$
are assumed to be well mixed within the mixed layer. The first two terms on the right-hand side of
equation (4) account for the effect of nutrient and light availability on autotrophic growth rates,
and they are hereafter defined as follows for simplicity:

$$N_m = \frac{N}{N + k_m^N} \qquad (5a)$$

$$I_m(z) = \frac{I(z)}{I(z) + k_m^I} \qquad (5b)$$

$I(z)$ is modeled as an exponential decay of PAR just beneath the water surface ($I_0$):

$$I(z) = I_0 \times e^{-K_I \times z} \qquad (6)$$

where $K_I$ is light attenuation coefficient which is assumed to be independent of depth in the mixed
layer.
As a first approximation, we assume that $HR(z)$ is proportional to $C$ as in previous studies
(Dutkiewicz et al., 2001; Huisman and Weissing, 1994; Rivkin and Legendre, 2001; Sverdrup,
1953; White et al., 1991):

$$HR(z) = r_{HR} \times C \qquad (7)$$

where $r_{HR}$ represents the intrinsic heterotrophic respiration rate which is assumed to be dependent
on temperature (see below), and independent of depth. In reality, $HR(z)$ is likely best modeled as
a function of the concentration of labile organic matter — an additional term could be included to
account for the relationship of total labile organic matter to $C$.
NCP integrated over the mixed layer ($NCP(0, MLD)$) can be derived from equations (3-7):
$$NCP(0, MLD) = NPP(0, MLD) - HR(0, MLD)$$
$$= \int_0^{MLD} NPP(z)dz - \int_0^{MLD} HR(z)dz$$
$$= N_m \times I_m(0, MLD) \times \mu_{max} \times C - r_{HR} \times MLD \times C \qquad (8)$$
The first term on the right side of equation (8) represents NPP integrated over the mixed layer
($NPP(0, MLD)$), which is equivalent to the product of $\int_0^{MLD} \mu(z)dz$ and $C$, where the former term
is modeled to be a function of $\mu_{max}$ conditioned by nutrient and light availability within the mixed
layer. $I_m(0, MLD)$ can be derived as follows:
$$I_m(0, MLD) = \int_0^{MLD} I_m(z)dz = -\frac{1}{K_I} \times ln\left(\frac{I_0 \times e^{-K_I \times MLD} + k_m^I}{I_0 + k_m^I}\right) \qquad (9)$$
NCP integrated over the mixed layer (equation (8)) is a bell-shaped function of MLD as depicted
in the schematic diagram of Figure 1(B).
**2.2. Net community production and phytoplankton biomass concentration**
As can be seen from equation (8), $NCP(0, MLD)$ is a direct function of $C$ because
$NPP(0, MLD)$ and $HR(0, MLD)$ are proportional to $C$. $NCP(0, MLD)$ is also an indirect function
of $C$ due its effect on light attenuation (i.e., $K_I$). The attenuation coefficient $K_I$ can be divided into
water and non-water components ($K_I = K_I^w + K_I^{nw}$) (Baker and Smith, 1982; Smith and Baker,
1978a; Smith and Baker, 1978b), where $K_I^{nw}$ is controlled by the concentrations of phytoplankton,
colored dissolved organic matter (CDOM), and non-algal particles (NAP). In the open ocean where
CDOM and NAP co-vary with phytoplankton (Morel and Prieur, 1977), $K_I$ can be related to $C$ as
follows:
$$K_I = K_I^w + k_c \times C \quad (10)$$

where $k_c$ is a function of the solar zenith angle, the specific absorption and backscattering
coefficients of phytoplankton, and the relationship between phytoplankton, CDOM, and NAP.
Because pure water and phytoplankton attenuate light, $K_I^w$ and $k_c$ should be greater than zero.
To calculate how $NCP(0, MLD)$ varies as a function of $C$, we examine its first $(\frac{dNCP(0,MLD)}{dC})$
and second $(\frac{d^2 NCP(0,MLD)}{dC^2})$ derivatives with respect to $C$ based on equations (8) and (10):
$$\frac{dNCP(0, MLD)}{dC}$$
$$= N_m \times \mu_{max} \times \frac{K_I^w \times I_m(0, MLD) + k_c \times C \times MLD \times I_m(MLD)}{K_I^w + k_c \times C} - r_{HR} \times MLD \quad (11)$$
$$\frac{d^2 NCP(0, MLD)}{dC^2} = N_m \times k_c \times \frac{\mu_{max}}{K_I}$$
$$\times \left\{ \frac{2 \times K_I^w}{K_I} \times \left( MLD \times I_m(MLD) - I_m(0, MLD) \right) - \frac{k_c \times C \times I_m(MLD)^2 \times MLD^2 \times k_m^I}{I_0 \times e^{-K_I \times MLD}} \right\} \quad (12)$$
when $MLD > 0$, $I_m(0, MLD) > MLD \times I_m(MLD)$:
$$I_m(0, MLD) = \int_0^{MLD} \frac{I_0 \times e^{-K_I \times z}}{I_0 \times e^{-K_I \times z} + k_m^I} dz$$
$$> \int_0^{MLD} \frac{I_0 \times e^{-K_I \times MLD}}{I_0 \times e^{-K_I \times MLD} + k_m^I} dz = MLD \times I_m(MLD) \quad (13)$$
The detailed derivation of equations (11-12) can be found in the supplementary material.
Substituting the inequality (13) into equation (12) gives $\frac{d^2 NCP(0,MLD)}{dC^2} < 0$, which suggests that
$\frac{dNCP(0,MLD)}{dC}$ decreases with increasing $C$. Because increasing $C$ decreases light availability due to
shelf-shading, $NPP(0, MLD)$ saturates with increasing $C$. Thus, $NCP(0, MLD)$ will reach an
asymptote of $\lim_{C \to \infty} \left( \frac{dNCP(0,MLD)}{dC} \right) = -r_{HR} \times MLD < 0$, because $HR(0, MLD)$ linearly increases
with increasing $C$ while $NPP(0, MLD)$ plateaus (Figure 2). Additionally, because $NCP(0, MLD)$
must be nil when there is no autotrophic biomass $(NCP(0, MLD)|_{C=0} = 0)$, $\lim_{C \to 0} \left( \frac{dNCP(0,MLD)}{dC} \right)$ must
be greater than zero, otherwise the ecosystem would be net heterotrophic which is unachievable
without an allochthonous source of organic matter. $\lim_{C \to 0} \left( \frac{dNCP(0,MLD)}{dC} \right) > 0$ and $\lim_{C \to \infty} \left( \frac{dNCP(0,MLD)}{dC} \right) =$
$-r_{HR} \times MLD < 0$ suggest the existence of $\frac{dNCP(0,MLD)}{dC}\bigg|_{C=C^*} = 0$ where $C^*$ corresponds to an
autotrophic biomass concentration which maximizes $NCP(0, MLD)$ (i.e., $NCP^*$).
The dependence of $NCP(0, MLD)$ on $C$ can be conceptually understood in the following way.
Given a water column with sufficient nutrients, the critical depth $Z_c$ and compensation depth $Z_p$
are expected to shoal as $C$ increases. When $C$ is low, $NCP(0, MLD)$ increases with $C$ because of
its greater impact on $NPP(0, MLD)$ than on $HR(0, MLD)$. As $C$ further increases, the increase in
$NPP(0, MLD)$ with $C$ slows because of light attenuation (i.e., $K_I$). There is therefore a $C^*$ which
maximizes the difference between $NPP(0, MLD)$ and $HR(0, MLD)$ leading to $NCP^*$ (Figure 2).
Beyond this point $(C^*)$, further increasing $C$ will cause self-shading and limit photosynthesis in the
deep part of the mixed layer, as a result decreasing $NCP(0, MLD)$. Beyond a critical biomass ($C_c$),
the ecosystem becomes net heterotrophic. Without an allochtonous source of organic carbon, this
is only transiently sustainable.
**2.3. Mixed layer depth and compensation depth**
By definition, if $NCP(MLD)$ is smaller than zero (i.e., net heterotrophy at the bottom of the
mixed layer), the MLD must be deeper than $Z_p$ $(MLD > Z_p)$ (and vice versa). To determine the
sign of $NCP(MLD)$, we substitute inequality (13) into equation (11). According to the inequality
presented in equation (13), $\frac{K_I^W \times I_m(0,MLD) + k_c \times C \times MLD \times I_m(MLD)}{K_I^W + k_c \times C}$ in equation (11) must be greater than
$\frac{K_I^W \times MLD \times I_m(MLD) + k_c \times C \times MLD \times I_m(MLD)}{K_I^W + k_c \times C}$ (which is equal to $MLD \times I_m(MLD)$ ). After simple
rearrangements, the substitution of inequality (13) into equation (11) leads to:
$$\frac{dNCP(0, MLD)}{dC}$$
$$> MLD \times (N_m \times I_m(MLD) \times \mu_{max} - r_{HR}) = \frac{MLD}{C} \times NCP(MLD) \qquad (14)$$
The inequality in equation (14) in turn suggests that when $NCP(0, MLD)$ is maximized
($\frac{dNCP(0,MLD)}{dC} = 0$), $NCP(MLD)$ is negative (net heterotrophic) and hence the MLD is deeper than
$Z_p$ ($MLD > Z_p$). This counterintuitive result is attributable both to the uneven distribution of light
availability in the water column (equation (13)) and to water which absorbs light but does not
contribute to biomass accumulation. When the mixed layer is at the $Z_p$, a slight increase in $C$ will
leads to negative $NCP(MLD)$ due to decreasing light availability at the base of mixed layer, but
will increase NCP higher in the water column because of the increase in biomass. The increase in
NCP in the shallow parts of the mixed layer therefore overcompensates for the net heterotrophy at
the bottom of the mixed layer, thus maximizing the depth-integrated NCP. If light were uniformly
distributed in the water column (i.e., $I_m(0, MLD) = MLD \times I_m(MLD)$) and if water did not
attenuate light ($K_I^W = 0$ in equation (11)), $MLD = Z_p$ would maximize $NCP(0, MLD)$, which is
consistent with Huisman and Weissing (1994). We note that in equation (14) the NCP profile
($NCP(z)$) varies with increasing $C$, which is different from what is conceptually presented in
Figure 1. The depth-integrated NCP in Figure 1 maximizes at the compensation depth because the
NCP profile ($NCP(z)$) is assumed to be invariant.
**2.4. An upper bound on carbon export**
Equations (11-13) delineate the conditions for an upper bound on carbon export ($NCP^*$). In
order to simplify the relationship of $NCP^*$ to MLD and temperature, we approximate $I_m(0, MLD)$:
$$I_m(0, MLD) = -\frac{1}{K_I} \times ln\left(1 + \frac{I_0}{I_0 + k_m^I} \times (e^{-K_I \times MLD} - 1)\right)$$

$$\approx -\frac{1}{K_I} \times ln(1 - I_m(0)) \qquad (15)$$

where $I_m(0) = \frac{I_0}{I_0 + k_m^I}$. Based on equation (15), $NCP(0, MLD)$ in equation (8) can be approximated
as:
$$NCP(0, MLD) = C \times MLD \times \left(\frac{1}{K_I \times MLD} \times \mu^* - r_{HR}\right) \qquad (16)$$

where $\mu^* = -ln(1 - I_m(0)) \times N_m \times \mu_{max}$. To evaluate the approximation accuracy of equation
(15), we compare the upper bounds estimated from equation (16) and the original model (equations
(8-10)). Our comparison suggests that the approximation of equation (15) is accurate for the
estimation of $NCP^*$ under most conditions (Figure 3).

We first need to derive the $C^*$ which maximizes $NCP(0, MLD)$ (i.e., $NCP^*$) in equation (16).

$C^*$ can be solved from the first derivative of $NCP(0, MLD)$ in equation (16) with respect to $C$:
$$\left.\frac{dNCP(0, MLD)}{dC}\right|_{NCP(0,MLD)=NCP^*} = \mu^* \times \frac{K_I^w}{(k_c \times C^* + K_I^w)^2} - MLD \times r_{HR} = 0 \qquad (17)$$


and therefore:
$$C^* = \frac{1}{k_c} \times \left[-K_I^w + \sqrt{\frac{\mu^* \times K_I^w}{MLD \times r_{HR}}}\right] \qquad (18)$$

Equation (18) decreases with MLD. As $C^*$ is positive ($C^* \geq 0$) and cannot go to infinity ($C^* \leq$
$C_{max}^*$), MLD should satisfy $MLD_{C_{max}^*} \leq MLD \leq \frac{\mu^*}{r_{HR} \times K_I^w}$, where $MLD_{C_{max}^*}$ represents the MLD
corresponding to the maximum achievable autotroph's biomass concentration ($C_{max}^*$) in the
surface ocean. The $NCP^*$ model for $0 \leq MLD < MLD_{C_{max}^*}$ is not discussed here, because we do
not have data with very shallow MLD to constrain and evaluate the model. The derivation of the
model is however presented in the supplementary material. Substituting $C^*$ from equation (18) into
equation (16):

$$\sqrt{NCP^*} = a_2 \times \sqrt{-ln(1 - I_m(0))} + a_1 \times \sqrt{MLD} \qquad (19)$$

where $a_1 = -\sqrt{\frac{K_I^W \times r_{HR}}{k_c}}$ and $a_2 = \sqrt{\frac{N_m \times \mu_{max}}{k_c}}$. Constants $a_1$ and $a_2$ are functions of $r_{HR}$ and $\mu_{max}$,
respectively, which are generally modeled to increase with temperature ($T$) (Eppley, 1972; Rivkin
and Legendre, 2001):

$$\mu_{max} = \mu_{max}^0 \times e^{P_t \times T} \qquad (20a)$$

$$r_{HR} = r_{HR}^0 \times e^{B_t \times T} \qquad (20b)$$

where $P_t$ and $B_t$ are constants; and $\mu_{max}^0$ and $r_{HR}^0$ are maximum growth rate and heterotrophic
respiration ratio for $T = 0$ ºC, respectively. $P_t$ is commonly assumed to equal 0.0663 (Eppley,
1972). Substituting equations (20a) and (20b) into equation (19) yields:

$$\sqrt{NCP^*} = a_4 \times \sqrt{e^{P_t \times T}} \times \sqrt{-ln(1 - I_m(0))} + a_3 \times \sqrt{e^{B_t \times T}} \times \sqrt{MLD} \qquad (21)$$

where $a_3 = -\sqrt{\frac{r_{HR}^0 \times K_I^W}{k_c}}$ and $a_4 = \sqrt{\frac{\mu_{max}^0 \times N_m}{k_c}}$.
**2.5. Comparison to observations**
**2.5.1 Data products**

We assess the performance of our modeled upper bound on carbon export using a global dataset

of MLD, PAR, sea surface temperature (SST), $O_2$/Ar-derived NCP, and export production derived
from sediment traps and [234]Th (see supplementary material). MLD was derived from global Argo
profiles (Global Ocean Data Assimilation Experiment; http://www.usgodae.org/) and CTD casts
(National Oceanographic Data Center; https://www.nodc.noaa.gov/). PAR was downloaded from

the NASA ocean color website (https://oceancolor.gsfc.nasa.gov/). The NCP estimates are based

the NASA ocean color website (https://oceancolor.gsfc.nasa.gov/). The NCP estimates are based
on a compilation of O₂/Ar measurements from Li and Cassar (2016), Li et al. (2016), Shadwick et
al. (2015), and Martin et al. (2013). The POC export production estimates were obtained from the
recently compiled dataset of Mouw et al. (2016). These estimates were adjusted to reflect a flux at
the base of mixed layer using the Martin curve of organic carbon attenuation with depth (Martin
et al., 1987). The constants $k_c$ and $K_I^w$ in equation (10) were derived assuming a carbon to
chlorophyll $a$ ratio of 90 (Arrigo et al., 2008) and an empirical linear relationship between $K_I$ and
chlorophyll $a$ concentration (see Figure S3), calculated based on the NOMAD dataset (Werdell
and Bailey, 2005). $k_m^I$ was set at 4.1 Einstein m$^{-2}$ d$^{-1}$ following Behrenfeld and Falkowski (1997).
In our estimation of the upper bound on carbon export, we set $N_m$ to 1 in the $NCP^*$ calculations.
**2.5.2 Results and discussion**
Overall, we find that $NCP^*$ calculated using published parameters (Table 2) does a good job
of enveloping carbon export observations reported in the literature (Figure 4(A)). Samples on the
$NCP^*$ envelope (upper bound) are likely regulated by light availability. Conversely, points below
the upper bound may be nutrient limited. As expected, $NCP^*$ increases with $\mu_{max}$ and decreases
with $r_{HR}$. Model parameters $a_1 = -1.78$ and $a_2 = 14.75$ (equation (19)) provide the best fit to the
upper bound of O₂/Ar-NCP as a function of MLD. When compared to parameters available in the
literature (Table 2), we find that the best fit to our modeled upper bound is using $\mu_{max}$ and $r_{HR}$ of
1.2 d$^{-1}$ and 0.2 d$^{-1}$, respectively. When accounting for the effect of $T$ on $\mu_{max}$ and $r_{HR}$, model
constants $a_3 = -1.53$ and $a_4 = 13.39$ (equation (21)) best fit the upper bound on O₂/Ar-NCP,
SST and MLD observations.
Our results show that $NCP^*$ decreases faster with increasing MLD in warmer waters (Figures
4(B) and 4(C)), because the term $a_3 \times \sqrt{e^{B_t \times T}}$ in equation (21) is negative and negatively
correlated to $T$. This temperature effect contributes to part of the relationship between export
production and MLD in Figure 4(A). Interestingly, $NCP^*$ increases with $T$ in colder waters and
shallow mixed layers (Figure 4(C)). This is because $NCP^*$ reflects the balance between
productivity ($a_4 \times \sqrt{e^{P_t \times T}} \times \sqrt{-\ln(1 - I_m(0))}$) and heterotrophic respiration ($a_3 \times \sqrt{e^{B_t \times T}} \times$
$\sqrt{MLD}$ ). In a shallow cold mixed layer, the change in productivity with $T$
($\frac{d\left(a_4 \times \sqrt{e^{P_t \times T}} \times \sqrt{-\ln(1 - I_m(0))}\right)}{dT} = \frac{P_t}{2} \times a_4 \times \sqrt{e^{P_t \times T}} \times \sqrt{-\ln(1 - I_m(0))}$) is greater than that of
heterotrophic respiration ($\frac{d\left(a_3 \times \sqrt{e^{B_t \times T}} \times \sqrt{MLD}\right)}{dT} = \frac{B_t}{2} \times a_3 \times \sqrt{e^{B_t \times T}} \times \sqrt{MLD}$). These results could
explain part of the variability in the relationship between NCP and SST reported in previous studies
(Li and Cassar, 2016). Our $NCP^*$ model does not perform as well in warmer deep mixed layers,
where high variability in export ratio maxima have also been reported (Cael and Follows, 2016).
This may stem from uncertainties in observations, the differing relationship between $T$, $\mu_{max}$ , and
$r_{HR}$ at high temperature, and/or violations of our assumptions (see caveats and limitations).
Several recent studies have explored the relationship of NCP to oceanic parameters based on
various statistical approaches (Cassar et al., 2015; Chang et al., 2014; Huang et al., 2012; Li and
Cassar, 2016; Li et al., 2016). Our model can shed some light into the mechanisms driving some
of these patterns. To that end, we substitute equation (9) into equation (8):
$$NCP(0, MLD) = C \times MLD \times \left(-\frac{N_m \times \mu_{max}}{K_I \times MLD} \times \ln\left(\frac{I_0 \times e^{-K_I \times MLD} + k_m^I}{I_0 + k_m^I}\right) - r_{HR}\right) \quad (22)$$
Rearranging equation (22):
$$NCP_B = \frac{NCP(0, MLD)}{C \times MLD} = -\frac{\ln\left(\frac{I_0 \times e^{-K_I \times MLD} + k_m^I}{I_0 + k_m^I}\right)}{I_0 \times (1 - e^{-K_I \times MLD})} \times N_m \times \mu_{max} \times PAR_{ML} - r_{HR} \quad (23)$$
where $NCP_B$ is the biomass-normalized volumetric NCP, $PAR_{ML}$ is the average PAR in the mixed
layer ($PAR_{ML} = \frac{1-e^{-K_I \times MLD}}{K_I \times MLD} \times I_0$), and $-\frac{ln\left(\frac{I_0 \times e^{-K_I \times MLD}+k_m^I}{I_0+k_m^I}\right)}{I_0 \times (1-e^{-K_I \times MLD})} \times N_m \times \mu_{max}$ and $-r_{HR}$ correspond
to the slope and offset, respectively. The scatter in the relationship between chlorophyll-
normalized volumetric NCP and $PAR_{ML}$, as reported in previous studies (Bender et al., 2016), can
likely be explained by the effect of temperature and the availability of nutrient and light (among
other properties) on the slope and offset of equation (23). Equation (22) can also be reorganized to
assess how environmental conditions may impact the export ratio ($ef$):
$$ef = \frac{NCP(0, MLD)}{NPP(0, MLD)} = 1 - \frac{K_I \times MLD}{-ln\left(\frac{I_0 \times e^{-K_I \times MLD} + k_m^I}{I_0 + k_m^I}\right)} \times \frac{1}{N_m} \times \frac{r_{HR}}{\mu_{max}} \quad (24)$$

where $\frac{r_{HR}}{\mu_{max}}$ is proportional to $e^{(B_t-P_t) \times T}$. Equation (24) is consistent with multiple studies which
predict decreasing $ef$ with increasing temperature (Cael and Follows, 2016; Dunne et al., 2005;
Henson et al., 2011; Laws et al., 2000; Li and Cassar, 2016). In fact, equation (5) of Cael and
Follows (2016) can easily be derived from equation (24) (see supplementary material). Equation
(24) also highlights that a multitude of factors may confound the dependence of $ef$ on temperature
(including varying MLD, light attenuation, and availability of nutrient and light). This again may
explain some of the conflicting observations recently reported in the literature (e.g., Maiti et al.
(2013)), where the effect of temperature may be masked by changes in community composition
(Britten et al., 2017; Henson et al., 2015). One therefore needs to account or correct for the
multitude of confounding factors when predicting the effect of a given environmental condition
(e.g., temperature, mineral ballast, and NPP) on the export ratio.
**3. Spatial distribution of the upper bound on carbon export**
We estimate the global distribution of the upper bound of carbon export using equation (19)
and climatological monthly MLD and PAR. In general, $NCP^*$ is high in low latitudes and low in
the North Atlantic and Antarctic Circumpolar Current (ACC) in the Southern Ocean (Figure 5(A)).
As expected, this spatial pattern is controlled by MLD (see Figure S1). Satellite-derived estimates
of NCP (Li and Cassar, 2016) are approximately 10% of global $NCP^*$, reflecting the high degree
of nutrient limitation in the oceans. We also derive a global $NCP^*$ map using equation (21), and
find that the global $NCP^*$ estimate is very sensitive to the temperature dependence of $r_{HR}$. For
example, decreasing the $B_t$ in $r_{HR} = r_{HR}^0 \times e^{B_t \times T}$ from 0.11 to 0.08 (as used in Rivkin and
Legendre (2001) and López-Urrutia et al. (2006)) increases the global $NCP^*$ budget by a factor of
2.4. Large differences in $NCP^*$ in low-latitudes in great part explain this change. In light of the
large uncertainties in the relationship between $r_{HR}$ and $T$ (Cael and Follows, 2016; López-Urrutia
et al., 2006), we hereafter only discuss $NCP^*$ estimates derived from equation (19).
To estimate how close export production is to its upper bound, we calculate the ratio of export
production to $NCP^*$ ($f_{pt}$). Low $f_{pt}$ regimes represent ecosystems likely regulated by nutrient
availability (i.e., ecosystems that have not reached their full export potential based on MLD and
surface PAR). As expected, low latitude and subtropical regions have low $f_{pt}$ (Figure 5(B)). High
$f_{pt}$ regimes represent ecosystems which have reached their full light potential, and are therefore
less likely to respond to nutrient addition because of light limitation (e.g., North Atlantic and ACC
(Figure 5(B))). In these regions, especially the subantarctic region, $f_{pt}$ is high in the spring (Figure
5(C)) and decreases in the summer (Figure 5(D)), suggesting that export production is likely co-
limited by nutrient and light availability. This may in part explain the lower response to iron
fertilization in the subantarctic region where substantial increases in surface chlorophyll were only
observed in regions with shallower mixed layers (Boyd et al., 2007; Boyd et al., 2000; de Baar et
al., 2005).

Also shown in Figure 5 are the biological pump efficiency and export ratio $ef$ (panels 5E and

5F, respectively). These various proxies reflect different components of the biological pump.
Whereas $f_{pt}$ reflects the export potential based on current MLD and light availability, the
biological pump efficiency reflects the potential as derived from nutrient distribution in the oceans,
estimated from the extent of nutrient removal from the surface ocean (Sarmiento and Gruber, 2006)
or the proportion of regenerated nutrients at depth (Ito and Follows, 2005). A revised estimate of
the global biological pump efficiency, estimated based on the proportion of regenerated to total
nutrients (preformed + regenerated) at depth is around 30-35% (Duteil et al., 2013). The $ef$ ratio
on the other hand describes how much of production is exported as opposed to recycled in the
surface (Dunne et al., 2005). The ultra-oligotrophic subtropical waters have a low export ratio, a
strong biological pump efficiency with exhaustion of nutrients at the ocean surface, and therefore
have not reached their full light potential (low $f_{pt}$) because of the strong stratification and nutrient
limitation. The seasonal pattern of $f_{pt}$ in the subantarctic region suggests that the low biological
pump efficiency is the result of light limitation in the austral spring and nutrient (likely Fe) and
light limitation in the austral summer.
**4. Caveats and limitations**

There are a multitude of uncertainties, simplifications, and approximations in our model and

field observations. Among others:

• In our study, we used a model which builds on Sverdrup's critical depth hypothesis. There

are competing hypotheses to explain phytoplankton bloom phenology (timing and

intensity), including the "dilution recoupling hypothesis" or "disturbance recovery

hypothesis" (Behrenfeld, 2010; Boss and Behrenfeld, 2010) and "critical turbulence
hypothesis" (Brody and Lozier, 2015; Huisman et al., 1999; Taylor and Ferrari, 2011). In
the case of top-down control, any respiratory grazing loss not accounted for by our loss
term would behave as a system not reaching its full light potential (NCP*). Conversely,
any grazing loss associated with export (e.g., rapidly sinking fecal pellets and other
zooplankton-mediated export pathways) would minimize respiratory losses thereby
bringing NCP closer to its upper bound based on light-availability. These opposing effects
are beyond the scope of this study, but could be modeled, especially as we learn more about
their impacts on carbon fluxes through new efforts such as NASA's EXPORTS program
(Siegel et al., 2016). See also the point below on mixing vs. mixed layer depth.
• Phytoplankton biomass concentration ($C$) may vary with depth in the mixed layer,
especially for water columns experiencing varying degrees of turbulent mixing. In addition,
MLD is not always the best proxy of light availability with mixing layer in some cases
deviating from the mixed layer (Franks, 2015; Huisman et al., 1999). The factors defining
the MLD also vary in different oceanic regions.
• For simplicity, we model the dependence of photosynthesis on irradiance assuming
Michaelis-Menten kinetics, which does not account for photoinhibition. More accurate
models can be found in other studies (Platt et al., 1980). Due to optional absorption, $K_I$
also varies with depth in the mixed layer. Additionally, the linear relationship between $K_I$
and $C$ is influenced by CDOM, NAP, and other environmental factors (e.g., solar zenith
angle) (Gordon, 1989).
• $\mu_{max}$ and $r_{HR}$ are influenced by environmental factors other than temperature, including
community structure (Chen and Laws, 2017), and may vary with depth within the mixed

layer (Smetacek and Passow, 1990). For these reasons, the equations relating $\mu_{max}$ and $r_{HR}$

(i.e., $B_t$ and $P_t$) to temperature also carry significant uncertainties (Bissinger et al., 2008;

Edwards et al., 2016; Kremer et al., 2017; López-Urrutia and Morán, 2007; Rivkin and

Legendre, 2001) which impacts our estimates of the upper bound on carbon export,

especially in warmer regions. As in other recent studies (Cael and Follows, 2016; Cael et

al., 2017; Dutkiewicz et al., 2001; Gong et al., 2015; Gong et al., 2017; Huisman et al.,

2006; Taylor and Ferrari, 2011), we model heterotrophic respiration to vary in proportion

to phytoplankton concentration. The model could be further improved by explicitly

including the concentration of heterotrophs. See point above on the grazing effect on export

with regards to $r_{HR}$.

• NCP may underestimate export production when accompanied by a decrease in the

inventory of organic matter in the mixed layer (see introduction and equation (2)).

• Our field observations are limited, mostly focusing on the spring and summer seasons, and

harbor significant uncertainties. For example, deep mixed layers can bias the $O_2/Ar$ method

low if entrainment of deeper waters brings low $O_2$ into the mixed layer. Descriptions of

these uncertainties are presented in other studies (Bender et al., 2011; Cassar et al., 2014;

Jonsson et al., 2013).

• Finally, our study is only relevant to the mixed layer. It does not account for productivity

below the mixed layer, which can be important in some regions such as the subtropical

ocean.

**5. Conclusions**
In this study, we derived a mechanistic model of an upper bound on carbon export ($NCP^*$) based
on the metabolic balance between photosynthesis and respiration of the plankton community. The
upper bound is a positively skewed bell-shaped function of mixed layer depth (MLD). At low
temperatures, the upper bound decreases with temperature if mixed layers are deep, but increases
with temperature if mixed layers are shallow. We used this model to derive a global distribution
of an upper bound on carbon export as a function of MLD and surface PAR, which shows high
values in low latitudes and low values in high latitudes due to deep MLD. To examine how current
export production compares to this upper bound in the world's oceans, we calculated the ratio of
satellite export production estimates to the upper bound derived by our model. High ratios of export
production to $NCP^*$ in the North Atlantic and ACC indicate that export production in these regions
is likely co-limited by nutrient and light availability. Overall, our results may explain differences
in carbon export measured during past iron fertilization experiments (e.g., subantarctic and polar
regions), inform future iron fertilization experiments, help in the development of remotely-sensed
carbon export algorithms, and improve predictions of the response of marine ecosystems to a
changing climate.
**Acknowledgements**
We would like to acknowledge NASA GSFC for processing and distributing PAR and SST
products (http://oceancolor.gsfc.nasa.gov/). Global Argo temperature-salinity profiling floats were
downloaded from http://www.usgodae.org/. CTD casts were downloaded from National
Oceanographic Data Center (https://www.nodc.noaa.gov/). N.C. was supported by NSF OPP-
1043339. Z.L. was supported by a NASA Earth and Space Science Fellowship (Grant No.
NNX13AN85H). The authors thank three anonymous reviewers for their insightful comments.

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

  **Table 1**. Model symbols, abbreviations, and units

| Symbol | Description | Units |
|---|---|---|
| MLD | Mixed layer depth | m |
| $MLD_{C^*_{max}}$ | Maximum MLD corresponds to maximum achievable autotroph's biomass concentration | m |
| $z$ | Depth | m |
| $Z_c$ | Critical depth | m |
| $Z_p$ | Compensation depth | m |
| GPP(0,z) | Gross primary production | mmol C m$^{-2}$ d$^{-1}$ |
| NPP(z) | Net primary production at depth z | mmol C m$^{-3}$ d$^{-1}$ |
| NPP(0,z) | Net primary production above depth z | mmol C m$^{-2}$ d$^{-1}$ |
| NCP(z) | Net community production at depth z | mmol C m$^{-3}$ d$^{-1}$ |
| NCP(0,z) | Net community production above depth z | mmol C m$^{-2}$ d$^{-1}$ |
| HR(z) | Heterotrophic respiration at depth z | mmol C m$^{-3}$ d$^{-1}$ |
| HR(0,z) | Heterotrophic respiration above depth z | mmol C m$^{-2}$ d$^{-1}$ |
| $NCP^*$ | The maximum NCP for a given MLD (upper bound on carbon export) | mmol C m$^{-2}$ d$^{-1}$ |
| $NCP_B$ | NCP normalized to autotroph's biomass inventory in the mixed layer | d$^{-1}$ |
| $ef$ | Export ratio | unitless |
| $f_{pt}$ | Ratio of satellite export production estimates to the upper bound on carbon export | unitless |
| N | Nutrient concentration | mmol m$^{-3}$ |
| $k_m^N$ | Half-saturation constant for nutrient concentration | mmol m$^{-3}$ |
| $N_m$ | Nutrient effect on phytoplankton grow $N_m = \frac{N}{N+k_m^N}$ | unitless |
| PAR | Photosynthetically active radiation | Einstein m$^{-2}$ d$^{-1}$ |
| $I_0$ | Photosynthetically active radiation just beneath water surface | Einstein m$^{-2}$ d$^{-1}$ |
| $I(z)$ | Photosynthetically active radiation at depth z | Einstein m$^{-2}$ d$^{-1}$ |
| $k_m^I$ | Half-saturation constant for irradiance | Einstein m$^{-2}$ d$^{-1}$ |
| $I_m(z)$ | Light effect on phytoplankton grow at depth z, $I_m(z) = \frac{I(z)}{I(z)+k_m^I} = \frac{I_0 \times e^{-K_I \times z}}{I_0 \times e^{-K_I \times z}+k_m^I}$ | unitless |
| $I_m(0,z)$ | Integrated light effect on phytoplankton grow above depth z, $I_m(0,z) = -\frac{1}{K_I} \times ln\left(\frac{I_0 \times e^{-K_I \times z}+k_m^I}{I_0+k_m^I}\right)$ | unitless |
| $PAR_{ML}$ | Average PAR in the mixed layer ($PAR_{ML} = \frac{1-e^{-K_I \times MLD}}{K_I \times MLD} \times I_0$) | Einstein m$^{-2}$ d$^{-1}$ |
| $\mu$ | Phytoplankton growth rate | d$^{-1}$ |
| $\mu_{max}$ | Maximum phytoplankton growth rate | d$^{-1}$ |

| | | |
|---|---|---|
| $\mu_{max}^0$ | Maximum phytoplankton growth rate for $T = 0$ °C | $d^{-1}$ |
| $r_{HR}$ | Heterotrophic respiration ratio | $d^{-1}$ |
| $r_{HR}^0$ | Heterotrophic respiration ratio for $T = 0$ °C | $d^{-1}$ |
| $K_I$ | Light attenuation coefficient ($K_I = K_I^w + K_I^{nw}$) | $m^{-1}$ |
| $K_I^w$ | Light attenuation coefficient due to water | $m^{-1}$ |
| $K_I^{nw}$ | Light attenuation coefficient due to optically active components | $m^{-1}$ |
| $k_c$ | Specific attenuation coefficient for irradiance | $m^2$ mmol$^{-1}$ |
| $C$ | Phytoplankton biomass concentration | mmol m$^{-3}$ |
| $C^*$ | Phytoplankton biomass concentration that maximizes NCP | mmol m$^{-3}$ |
| $C_{max}^*$ | Maximum achievable autotroph's biomass concentration | mmol m$^{-3}$ |
| POC | Particulate organic carbon | mmol m$^{-3}$ |
| DOC | Dissolved organic carbon | mmol m$^{-3}$ |
| CDOM | Colored dissolved organic matter | $m^{-1}$ |
| NAP | Non-algal particles | mmol m$^{-3}$ |
| $T$ | Temperature | °C |
| $P_t$ | Temperature dependence for phytoplankton grow rate | °C$^{-1}$ |
| $B_t$ | Temperature dependence for heterotrophic respiration ratio | °C$^{-1}$ |
| $CO_2$ | Carbon dioxide | ppmv |



**Table 2**. Value or range of values with references for the parameters used in the model.

| Parameter | Range or value | Reference |
|---|---|---|
| $K_I^w$ | 0.09 | (Werdell and Bailey, 2005) |
| $k_c$ | 0.03 | (Werdell and Bailey, 2005) |
| Carbon to chlorophyll ratio | 90 | (Arrigo et al., 2008) |
| $k_m^I$ | 4.1 Einstein m$^{-2}$ d$^{-1}$ | (Behrenfeld and Falkowski, 1997) |
| $P_t$ | 0.0663 | (Eppley, 1972) |
| $B_t$ | 0.08 | (Rivkin and Legendre, 2001; López-Urrutia et al., 2006) |
| $\mu_{max}$ | 1 d$^{-1}$, 1.2 d$^{-1}$ | (Laws et al., 2000; Eppley, 1972) |
| $r_{HR}$ | 0.1 d$^{-1}$, 0.2 d$^{-1}$ | (Laws et al., 2000; Mitchell et al., 1991) |




**Figure 1.** Schematic diagram of depth-profiles of net community production (NCP), net primary
production (NPP), and heterotrophic respiration (HR). Yellow and black dots represent the
compensation and critical depths, respectively.
**Figure 2.** Relationship between net primary production (NPP), heterotrophic respiration (HR),
net community production (NCP), and phytoplankton biomass concentration (C) for a given
mixed layer depth (MLD). Hatched area in panel A represents NCP. The yellow dot represents
the maximal NCP (NCP*) obtainable for a given MLD, with the corresponding phytoplankton
biomass concentration (C*) denoted with a cyan dot. NCP on the right of the yellow dot
decreases with C due to self-shading. Black dot represents depth-integrated NCP =0 (i.e.,
NPP=HR), with the corresponding phytoplankton biomass concentration defined as critical
biomass ($C_c$) and denoted with a blue dot. Ecosystems on the left and right of this threshold are
net autotrophic and heterotrophic, respectively. The asymptote (dashed blue line) in panel B
represents a system dominated by heterotrophic respiration (i.e., NCP ≈ HR >> NPP).
**Figure 3**. Upper bounds derived using the original and approximated models. The upper bound
for the original model (equations (8-10)) is estimated through a non-linear optimization
approach. The upper bound for the approximated model is calculated analytically from equation
(19). The models use the constants listed in Table 2 and $I_m(0) = 0.9$. Decreasing $I_m(0)$ and
increasing $r_{HR}$ results in greater discrepancies  between the original and approximated models in
regions with shallow mixed layers.
**Figure 4.** Modeled upper bound on carbon export production compared to field observations as a
function of mixed layer depth (MLD) and sea surface temperature (SST). (A) The thick gray line
represents the upper bound fitted to the net community production (NCP) data. Dash-lines
represent the upper bounds calculated using parameters available in the literature (Table 2). (B)
NCP as a function of SST with isopleths of constant upper bounds color coded for MLD. NCP
observations are color coded with MLD. (C) Surface representing the envelope of the modeled
upper bound of carbon export production as a function of SST and MLD. Bars represent field
observations color coded with the ratio of NCP to the upper bound. Observations are based on
$^{234}$Th and sediment traps estimates of carbon export production and $O_2/Ar$-derived NCP. A
stoichiometric ratio of $O_2/C=1.4$ was used to convert NCP from $O_2$ to C units (Laws, 1991). To
account for the effect of PAR on export production, both MLD and carbon fluxes are normalized
to $-log(1 - I_m(0))$ (see equations (19) and (21)). The temperature dependence of $r_{HR}$ was
modeled as $r_{HR} = r_{HR}^0 \times e^{0.08 \times T}$.
**Figure 5.** (A) Modeled upper bound on carbon export derived from equation (19), (B-D) ratios
of satellite export production estimates to the upper bound on carbon export, (E) biological pump
efficiency calculated as the difference in nutrient concentrations between surface and depth,
normalized to nutrient concentrations at depth (Sarmiento and Gruber, 2006) (nitrate
concentration from World Ocean Atlas (https://www.nodc.noaa.gov/OC5/woa13/)), and (F)
export ratio derived from Dunne et al. (2005). Annual represents annually-integrated value.
Spring and summer represent average value in spring and summer, respectively. In the northern
hemisphere, spring and summer seasons are defined as March-May and June-August,
respectively. In the southern hemisphere, spring and summer seasons are defined as September-
November and December-February, respectively.

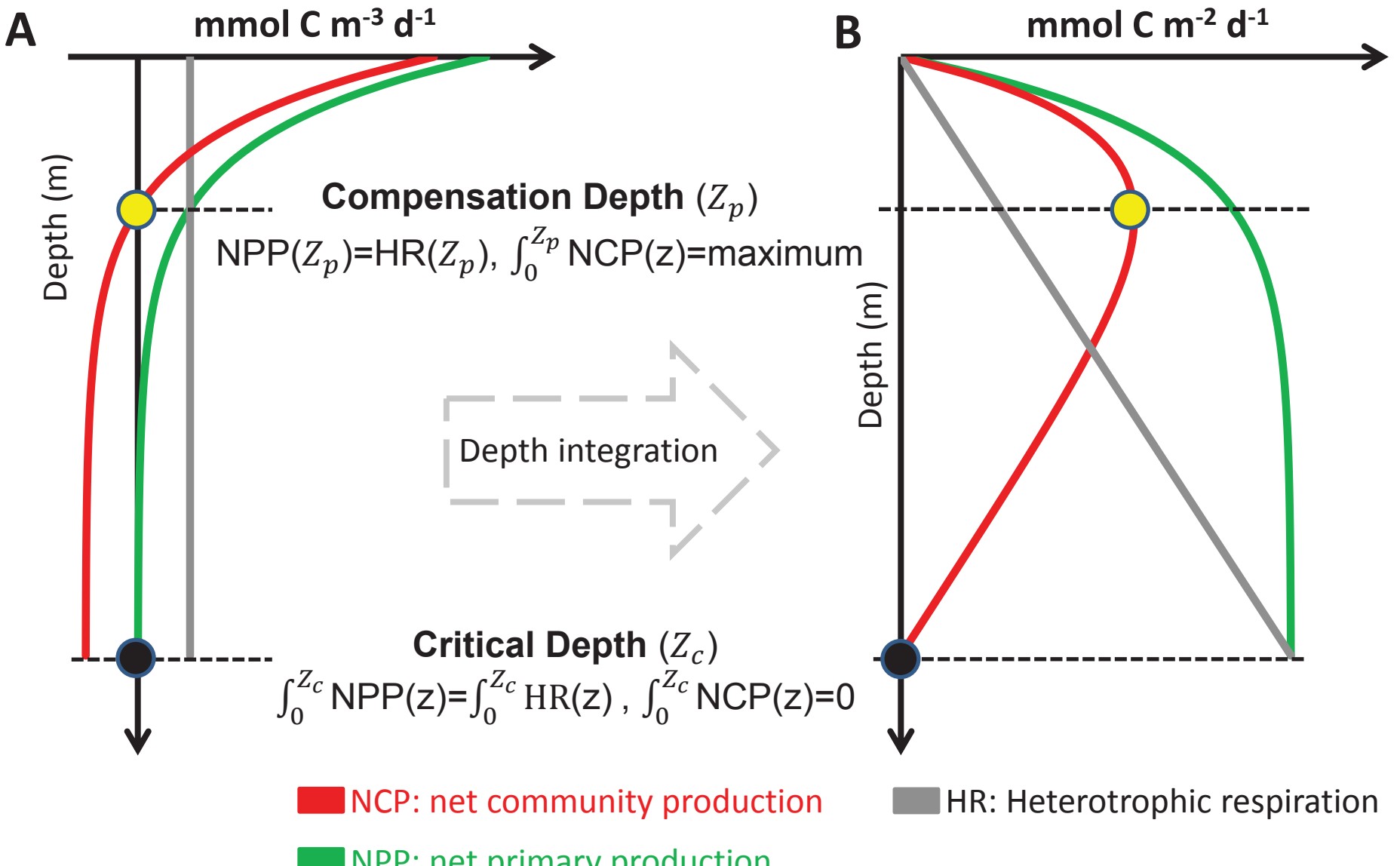

**A** mmol C m⁻³ d⁻¹

Depth (m)

**Compensation Depth** ($Z_p$)

$NPP(Z_p)=HR(Z_p)$, $\int_0^{Z_p} NCP(z)=$ maximum

Depth integration

**Critical Depth** ($Z_c$)

$\int_0^{Z_c} NPP(z)=\int_0^{Z_c} HR(z)$ , $\int_0^{Z_c} NCP(z)=0$

**B** mmol C m⁻² d⁻¹

Depth (m)

🟥 NCP: net community production     ⬛ HR: Heterotrophic respiration

🟩 NPP: net primary production

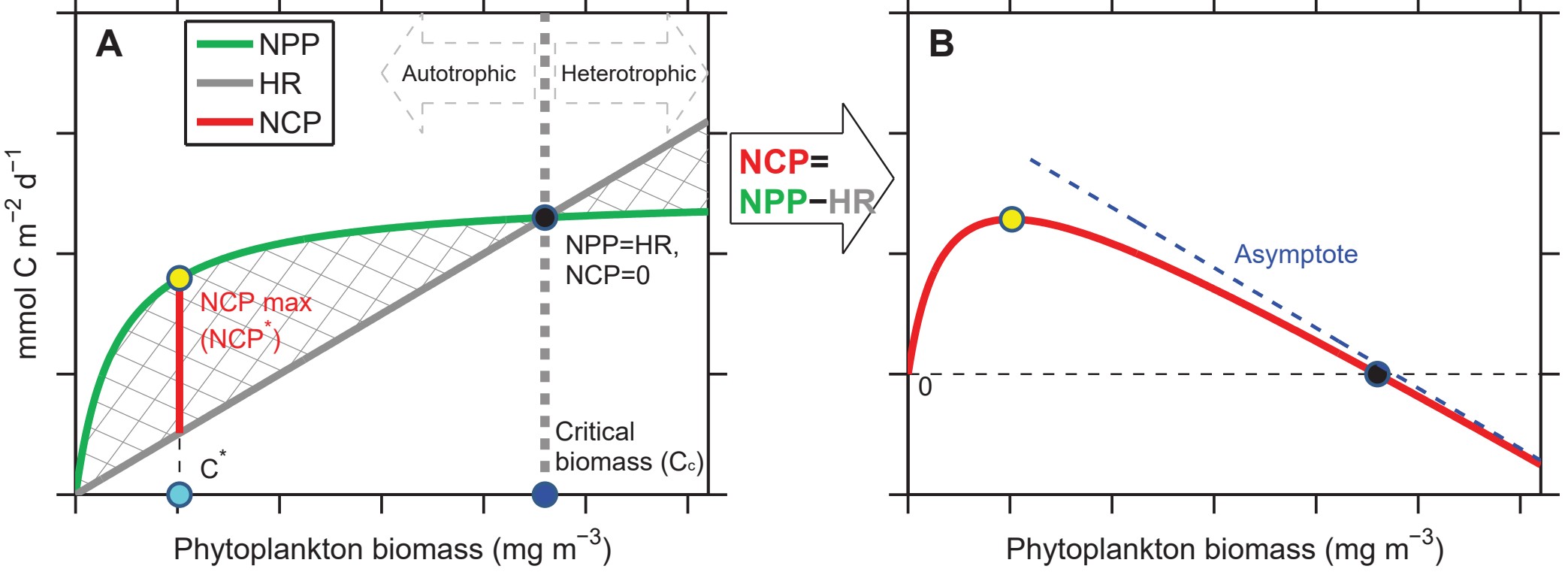

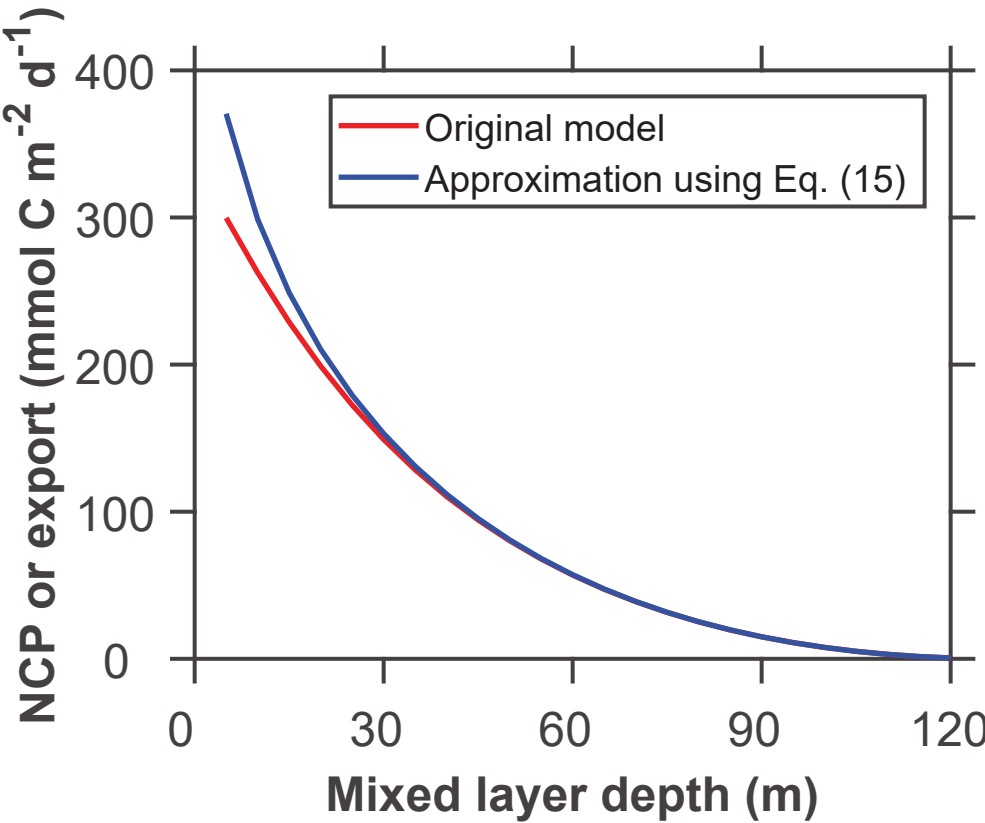

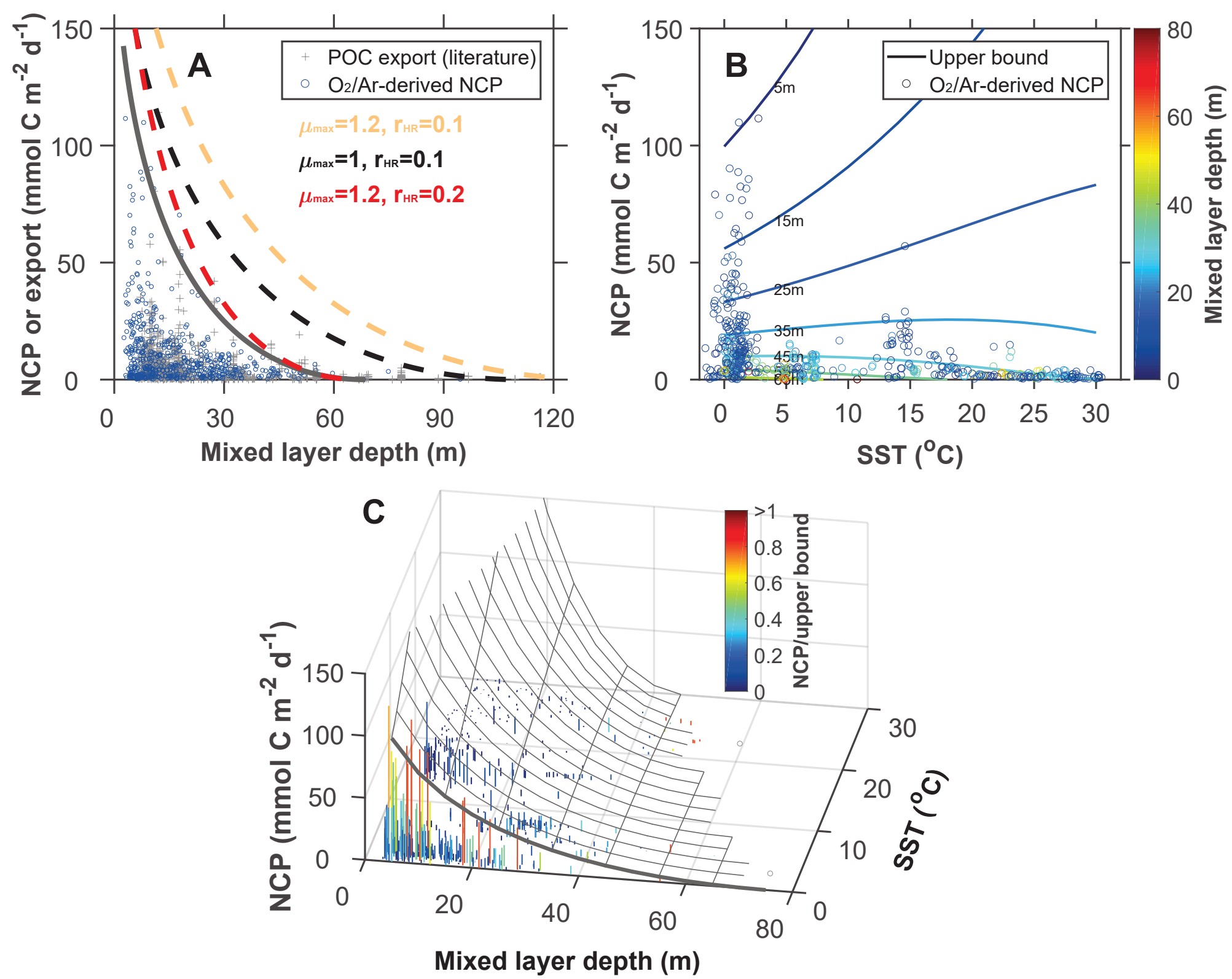

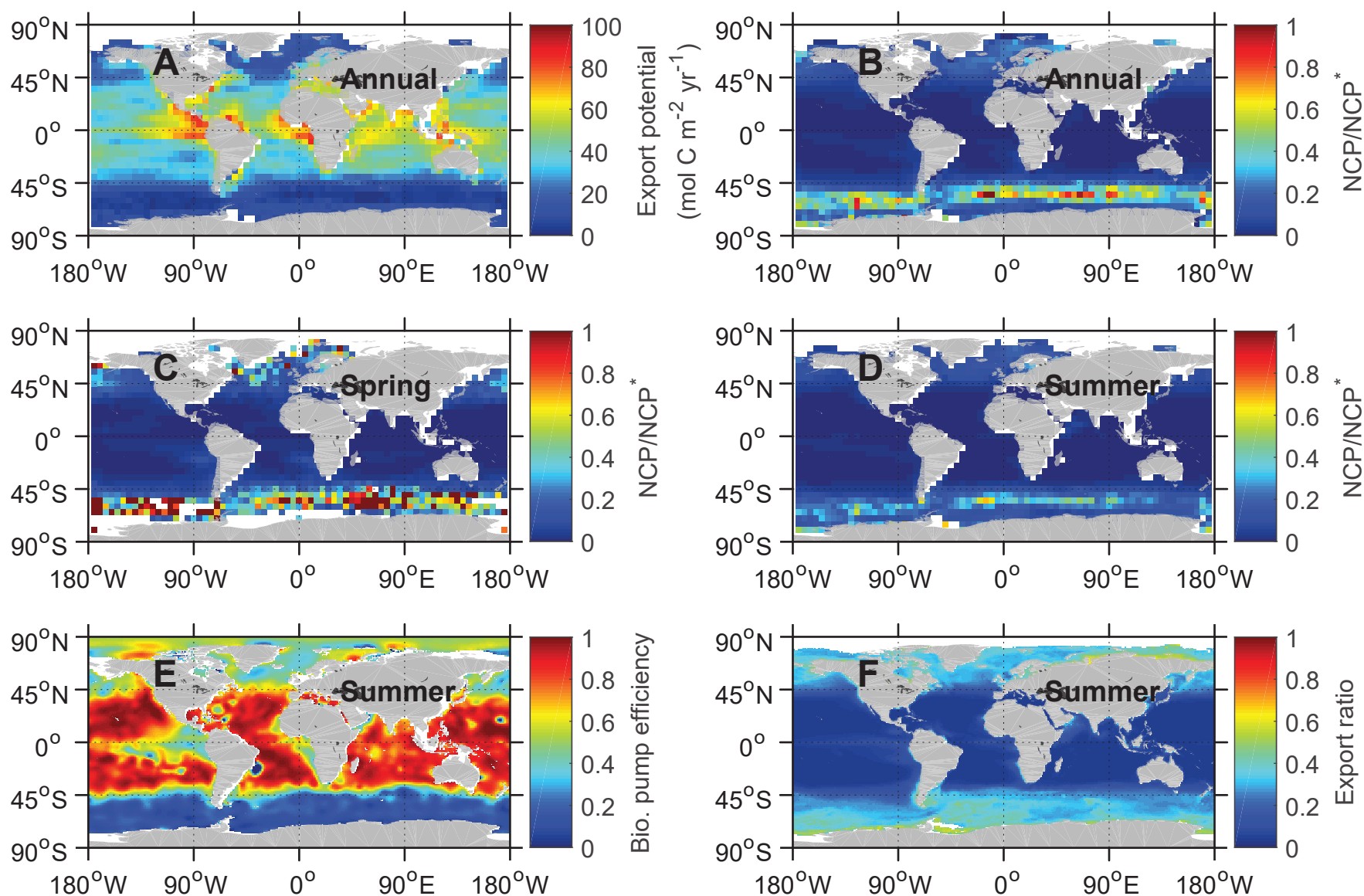