# Peer review of "A mechanistic model of an upper bound on oceanic carbon export as a 1 function of mixed layer depth and temperature 2 Zuchuan Li\*, Nicolas Cassar 3 Division of Earth and Ocean Sciences, Nicholas School of the Environment, Duke University, 4 5 Durham, North Carolina, USA 6 \* Corresponding author"

_Biogeosciences, 2017_

## Referee Comment (RC1) · Anonymous Referee #1 · 8 Jul 2017

The manuscript "A mechanistic model of an upper bound on oceanic carbon export as a function of mixed layer depth and temperature" by Li and Cassar applies the Sverdrup hypothesis to derive a simple maximum potential steady state primary production and carbon export estimate consistent with temperature and light limitation and mixed layer depth, and show that this maximum constraint is consistent with past estimates of carbon export. As such, the analysis seems incomplete in failing to describe what new insight the current theoretical constraint provides. Further, as the mathematical posing an equation for maximum possible export includes extremely simplified assumptions such as first-order herbivory that is constant with depth, the robustness of this constraint is impossible to quantify. Without demonstration that the existence of this

equation assists in the characterization of ocean ecosystems beyond what was available previously such as the ability to falsify otherwise seemingly reasonable hypotheses relating to primary production and carbon export, the contribution of the present manuscript to the scientific literature on these topics is unclear. I therefore recommend rejection of the present manuscript to encourage the authors to more fully develop the applications of their approach to demonstrate its usefulness.

---

## Referee Comment (RC2) · Anonymous Referee #2 · 12 Jul 2017

This is a nice, clearly written paper, based on an interesting idea and executed well.

The paper could be improved by clarifying the significance of the study somewhat. It may be very difficult ever to test or 'validate' this model properly. Yet, it is conceptually useful in some ways, e.g. the discussion about f_pt and nutrient limitation. The authors might want to discuss further, or clarify the existing discussion of, what the reader is supposed to have learned about the ocean.

There are several circumstances where the manuscript could be connected better to the literature. For instance, in line 30, there should be at least one reference for this sentence (good references should be easy to find from the reference list in Boyd, 2015)

- same for the next sentence. Dunne et al (2005) and Cael and Follows (2016) develop mechanistic models to which this paper is very directly related, yet these models are mentioned only in passing. It is worth mentioning that not everyone loves the Sverdrup model (Behrenfeld, 2010), though using it in this context is a nice idea. Some readers might also take issue with the sentence starting on line 31 - it's better to say 'export production is frequently assumed to be a function of' (e.g. Estapa et al, 2015), though the rest of the paragraph deals with this nicely.

It seems a bit ironic to compare this model, which is mechanistic, quite sophisticated, and carefully developed, with export data extrapolated using the Martin curve (an empirical parameterization) with a constant b-value. Granted, the model must be validated in some way, but the 'comparison to observations' subsection of the paper definitely appears to be its weak point.

Figures 3+4 are somewhat difficult to see/understand. The maps could be larger, and the axis limits could be chosen in a way to present the information more clearly.

Eq. 21 may be missing a normalizing constant - a proportionality (Eq. 20) is not the same as an equals sign. The values of $P_t$ and $B_t$ both merit a bit more discussion - both numbers have some associated uncertainty, do they not?

---

## Referee Comment (RC3) · Anonymous Referee #3 · 9 Aug 2017

This paper develops a theoretical estimate of the maximum NCP that can be produced by the ocean for a given season (surface light) and mixed layer depth. Differences between this theoretical maximum and observed levels can then be attributed to nutrient limitation. The model appears to explain the distribution of observations that show higher NCP values are achieved in shallower mixed layers. The contribution of the paper to the field could be better highlighted and some areas need justification to prove their methods are valid. With these additions and clarifications, the paper could be a substantial contribution to the field.

The paper would benefit from more motivation for the model at the start. The introduc-

tion is fairly short and general. The reader would be more eager to dive into all the details of the model if the need for this model and the questions that the authors hope to address with it were clearly laid out near the beginning of the paper. Figure 3 demonstrates that there are patterns in the observations that we should seek to explain, but this is only briefly introduced at the start of the paper. Figure 4 shows intuitive results, so here too the motivation to do the global analysis should be specifically stated.

A large proportion of export is potentially controlled by bloom dynamics as phytoplankton escape heterotrophic grazing control or not. The proposed model misses these dynamics by forcing heterotrophic respiration to be solely proportional to phytoplankton concentration, rather than also include heterotroph concentrations. Of course, this simplifies the model considerably. However, this simplification may render the results irrelevant since the model then does not approximate the real system closely enough. At the very least, the authors need to carefully argue that their model remains valid for the questions they wish to address despite this simplification of heterotrophic respiration. Such an argument is presently missing from the paper.

I would like to see more clarity about how the generalized conclusions of the model depend on choices for specific constants. For example, the discussion in the paragraph beginning on line 121 only holds where kc is significant. As kc goes toward zero, self-shading decreases and NPP will continuously increase as C increases. The text is not clear on whether the kc required to cause the self-shading induced decrease in dNCP/dC above a certain C is reasonable. The paper discusses specific values for some of these constants later in section 2.5, but it seems as though the values of these constants affect earlier conclusions as well.

The simplification in the last part of equation 15 appears to remove the dependence of average mixed layer irradiance on the depth of the mixed layer. Equation 16, based on this simplification, demonstrates that only the respiration term is now sensitive to the mixed layer depth (MLD cancels from the first term). This seems to run counter to all the previous arguments that MLD is important to integrated NPP values.

Lines 51-56: The discussion of attribution of these patterns seems too limited. Low NCP at high temperatures could be primarily a function of a tendency toward increased stratification and nutrient limitation in warm waters. Additionally, deep mixed layers can bias the O2/Ar method low if entrainment of deeper waters brings low oxygen into the mixed layer.

Line 82: "light" attenuation coefficient rather than "diffusion" attenuation coefficient?

Lines 113-120 and following paragraph: This section is unclear in places. Figure 2 could be actively discussed to demonstrate why dNCP/dC asymptotes at –r*MLD through comparison of the production and respiration terms on the right side of Figure 2a where the production term becomes stable. I spent a long time thinking about this, so the authors could really lead the reader through these arguments better. The text implies in places that dNCP/dC always decreases with increasing C (lines 113-114), but this is only the case at C larger than C*.

Lines 138-140: the statement here that integrated NCP is maximized when the MLD is below the compensation depth seems contrary to the schematic representation of the system in Figure 1a vs. 1b where the integrated NCP is maximized at the compensation depth.

Line 163: Why the MLD should satisfy the given conditions are not clear here until Line 171, where the authors state that they have chosen not to consider other possibilities.

Equations 20a and 20b: These are written as simple proportionalities here, but later treated as though the proportional sign is replaced with an equal sign. It seems like there should be an additional constant.

Section 2.5: Where specific values or ranges of values are chosen for model constants, it would be helpful to list these in the table defining notation.

Line 196: It's unclear why data could be below the theoretical line due to light limitation, when the theoretical line is specifically modeled to include light limitation.

Model-data differences are difficult to clearly discern in Figure 3b. Perhaps it would be useful to directly plot model-data differences in a third panel. That the NCP* model performs poorly in warm deep mixed layers (as stated on lines 210-211) cannot be clearly seen in the figure.

Line 281: The text discusses discrepancies between predicted and observed NCP*. However, only NCP can be observed, not NCP*.

---

## Author Comment (AC1) · 13 Sep 2017

Wednesday, September 13, 2017

Jack Middelburg
Editor
Biogeosciences

Dear Dr. Middelburg,

We would first like to thank you and the three reviewers for the careful examination of our manuscript and the insightful comments. We have taken into account these comments in the revised manuscript.

Below is a response to the reviewers' comments and the revised manuscript. Please do not hesitate to contact us should you have any additional questions or comments on our manuscript.

Sincerely,
Zuchuan Li

Division of Earth and Ocean Sciences
Nicholas School of the Environment
Duke University
email. zuchuan.li@duke.edu

**Reviewer 1:**

We thank the reviewer for his/her review of our manuscript. We disagree with his/her overall assessment. Below, we provide a response to the reviewer's two points:

- The reviewer's first comment states that our results "... show that this maximum constraint is consistent with past estimates of carbon export. As such, the analysis seems incomplete in failing to describe what new insight the current theoretical constraint provides"

As stated in the manuscript, our impetus for this study is to explain the recently reported field observations of an interesting relationship between export production proxies and mixed layer depths (Cassar et al., 2011; Eveleth et al., 2017; Tortell et al., 2015). Our theoretical considerations build on the qualitative description provided in these original studies.

We now further emphasize the key outcomes of our study in the introduction section of the manuscript, and enumerate them here: 1) the development of a mechanistic model of an upper bound on carbon export based on the metabolic balance of photosynthesis and respiration in the oceanic mixed layer, 2) using parameters available in the literature, the modeled upper bound envelopes field observations of export production estimated from $^{234}$Th and sediment traps and $O_2$/Ar-derived net community production, and 3) the model identifies regions of the Southern Ocean where carbon export is likely limited by light during part of the growing season. Our effort has significant implications for unraveling the influence of light and nutrients availability on carbon export production in the surface ocean (see Figure 5 of the revised manuscript), and for the development of models of export production based on satellite dataproducts.

Numerous recent modeling efforts have used simplified models to explore patterns in field observations. As an example, we refer the reviewer to the recent study of Cael and Follows (2016). In their study, the authors elegantly use "what is arguably the simplest mechanistic model" to explain the observed dependence of carbon export efficiency on temperature.

- The reviewer's second comment, related to the first, states that "...the mathematical posing an equation for maximum possible export includes extremely simplified assumptions such as first-order herbivory that is constant with depth..."

We, again, refer the reviewer to the multitude of recent modelling efforts which have used simplified equations for complex biogeochemical processes, including herbivory. Many recent studies use first-order kinetics for grazing losses and other assumptions (e.g., see Equation 1 in Cael and Follows, 2016; Cael et al., 2017; Dutkiewicz et al., 2001; Gong et al., 2015; Gong et al., 2017; Huisman et al., 2006; Taylor and Ferrari, 2011).

Most (if not all) of these recent studies also assume constant herbivory and biogeochemical properties with depth within the mixed layer (Cael and Follows, 2016; Cael et al., 2017; Dutkiewicz et al., 2001; Gong et al., 2015; Gong et al., 2017; Huisman et al., 2006; Siegel et al., 2014; Taylor and Ferrari, 2011). Nonetheless, we now further describe in the manuscript the limitations associated with Sverdrup's assumption of homogeneously mixed organisms or constant loss rates with depth within the mixed layer.

Overall, the reviewer's comments are unfounded in light of the fact that 1) many other *recently* published articles have used similar modeling approaches and equations, and 2) to the best of our knowledge, this is the first study to provide a theoretical constraint on an upper bound of carbon export fluxes as a function of light availability, mixed layer depth and temperature.

**References**

Cael, B. B., and M. J. Follows (2016), On the temperature dependence of oceanic export efficiency, Geophys Res Lett, 43(10), 5170-5175.

Cael B. B., K. Bisson, and M. J. Follows (2017), How have recent temperature changes affected the efficiency of ocean biological carbon export? Limnology and Oceanography Letters, DOI: 10.1002/lol2.10042

Cassar, N., P. J. DiFiore, B. A. Barnett, M. L. Bender, A. R. Bowie, B. Tilbrook, K. Petrou, K. J. Westwood, S. W. Wright, and D. Lefevre (2011), The influence of iron and light on net community production in the Subantarctic and Polar Frontal Zones, Biogeosciences, 8(2), 227-237.

Dutkiewicz, S., M. Follows, J. Marshall, and W. W. Gregg (2001), Interannual variability of phytoplankton abundances in the North Atlantic, Deep-Sea Res Pt Ii, 48(10), 2323-2344.

Eveleth, R., N. Cassar, R. M. Sherrell, H. Ducklow, M. Meredith, H. Venables, Y. Lin, and Z. Li (2017), Ice melt influence on summertime net community production along the Western Antarctic Peninsula, Deep Sea Research Part II, 137, 89-102.

Gong, X., J. Shi, H. W. Gao, and X. H. Yao (2015), Steady-state solutions for subsurface chlorophyll maximum in stratified water columns with a bell-shaped vertical profile of chlorophyll, Biogeosciences, 12, 905-919.

Gong, X., W. Jiang, L. Wang, H. Gao, E. Boss, X. Yao, S. Kao, and J. Shi (2017), Analytical solution of the nitracline with the evolution of subsurface chlorophyll maximum in stratified water columns, Biogeosciences, 14, 2371-2386.

Huisman, J., N. N. P. Thi, D. M. Karl, and B. Sommeijer (2006), Reduced mixing generates oscillations and chaos in the oceanic deep chlorophyll maximum, Nature, 439, 322-325.

Taylor, J. R., R. Ferrari (2011), Shutdown of turbulent convection as a new criterion for the onset of spring phytoplankton blooms, Limnol. Oceanogr., 56(6), 2293-2307.

Tortell, P. D., H. C. Bittig, A. Kortzinger, E. M. Jones, and M. Hoppema (2015), Biological and physical controls on $N_2$, $O_2$, and $CO_2$ distributions in contrasting Southern Ocean surface waters, Global Biogeochem Cy, 29(7), 994-1013.

**Reviewer 2:**

We thank the reviewer for his/her careful review of our manuscript. Below, we provide a response to the reviewer's comments which we think have significantly improved the quality of our manuscript:

Reviewer's comment: "This is a nice, clearly written paper, based on an interesting idea and executed well. The paper could be improved by clarifying the significance of the study somewhat. It may be very difficult ever to test or 'validate' this model properly. Yet, it is conceptually useful in some ways, e.g. the discussion about $f_{pt}$ and nutrient limitation. The authors might want to discuss further, or clarify the existing discussion of, what the reader is supposed to have learned about the ocean."

Following the reviewer's comment, we now describe at the end of the introduction some of the key outcomes of our study:

"In our study, we build upon Sverdrup (1953) and derive a mechanistic model of an upper bound on carbon export based on the metabolic balance of photosynthesis and respiration in the oceanic mixed layer, where the metabolic balance is derived from MLD, temperature, photosynthetically active radiation (PAR), phytoplankton maximum growth rate ($\mu_{max}$), and heterotrophic activity. Our approach is analogous to other efforts where mechanistic models were derived to predict proxies of carbon export (e.g., Dunne et al. (2005) and Cael and Follows (2016)). We compare our $NCP^*$ model to observations, and use this model in conjunction with satellite export production estimates to identify regions in the world's oceans where light may limit export production. Our key findings are that 1) using parameters available in the literature, the modeled upper bound envelopes field observations of export production estimated from [234]Th and sediment traps and $O_2$/Ar-derived net community production, and 2) the model identifies regions of the Southern Ocean where carbon export is likely limited by light during part of the growing season."

Reviewer's comment: "There are several circumstances where the manuscript could be connected better to the literature. For instance, in line 30, there should be at least one reference for this sentence (good references should be easy to find from the reference list in Boyd (2015) - same for the next sentence. Dunne et al (2005) and Cael and Follows (2016) develop mechanistic models to which this paper is very directly related, yet these models are mentioned only in passing."

Following the reviewer's recommendation, we now more explicitly make reference to the literature, including citations found in Boyd (2015). We agree with the reviewer that our effort is in the same vein as Dunne et al. (2005) and Cael and Follows (2016) and we now further emphasize the parallels in our approaches. The following references were added to the end of the first paragraph: "(Falkowski et al., 1998; Ito and Follows, 2005; Sigman and Boyle, 2000)." We also included references to Dunne et al. (2005) and Cael and Follows (2016) in the last paragraph of the introduction: "Our approach is analogous to other efforts where mechanistic models were derived to predict proxies of carbon export (e.g., Dunne et al. (2005) and Cael and Follows (2016))."

Reviewer's comment: "It is worth mentioning that not everyone loves the Sverdrup model (Behrenfeld, 2010), though using it in this context is a nice idea."

We now refer to the competing models of "dilution recoupling hypothesis" or "disturbance recovery hypothesis" and "critical turbulence hypothesis" in the section on "caveats and limitations" and cite the relevant literature:
"In our study, we used a model which builds on Sverdrup's critical depth hypothesis. There are competing hypotheses to explain phytoplankton bloom phenology (timing and intensity), including the "dilution recoupling hypothesis" or "disturbance recovery hypothesis" (Behrenfeld, 2010; Boss and Behrenfeld, 2010) and "critical turbulence hypothesis" (Brody and Lozier, 2015; Huisman et al., 1999; Taylor and Ferrari, 2011). In the case of top-down control, any respiratory grazing loss not accounted for by our loss term would behave as a system not reaching its full light potential (NCP*). Conversely, any grazing loss associated with export (e.g., rapidly sinking fecal pellets and other zooplankton-mediated export pathways) would minimize respiratory losses thereby bringing NCP closer to its upper bound based on light-availability. These opposing effects are beyond the scope of this study, but could be modeled, especially as we learn more about their impacts on carbon fluxes through new efforts such as NASA's EXPORTS program (Siegel et al., 2016). See also the point below on mixing vs. mixed layer depth."

Reviewer's comment: "Some readers might also take issue with the sentence starting on line 31 - it's better to say 'export production is frequently assumed to be a function of' (e.g. Estapa et al, 2015), though the rest of the paragraph deals with this nicely."

We have modified the sentence following the reviewer's comment to: "export production is frequently assumed to be a function of".

Reviewer's comment: "It seems a bit ironic to compare this model, which is mechanistic, quite sophisticated, and carefully developed, with export data extrapolated using the Martin curve (an empirical parameterization) with a constant b-value. Granted, the model must be validated in some way, but the 'comparison to observations' subsection of the paper definitely appears to be its weak point."

A study recently published shows that the fit of the Martin curve to observations is as good as more sophisticated parameterizations which account for the ballast effect (Gloege et al. 2017). However, we agree with the reviewer that using the Martin curve to extrapolate the carbon export observations to the base of the mixed layer introduces uncertainties. To circumvent this issue, we now also present a figure in the supplementary material which only includes biological carbon fluxes directly measured within the mixed layer:

[Figure]

**Figure S4**. Modeled upper bound on carbon export production compared to field observations as a function of mixed layer depth (MLD). Observations are based on $O_2$/Ar-derived net community production (NCP). To account for the effect of photosynthetically active radiation (PAR) on export production, both MLD and carbon fluxes are normalized to $-log(1 - I_m(0))$ (see equations (19) and (21)). The thick gray line represents the upper bound fitted to the NCP data. Dash-lines represent the upper bounds calculated using parameters available in the literature (Table 2). A stoichiometric ratio of $O_2$/C=1.4 was used to convert NCP from $O_2$ to C units (Laws, 1991).

Gloege, L., McKinley, G. A., Mouw, C. B., and Ciochetto, A. B.: Global evaluation of particulate organic carbon flux parameterizations and implications for atmospheric pCO₂, Global Biogeochemical Cycles, 2017.

Reviewer's comment: "Figures 3+4 are somewhat difficult to see/understand. The maps could be larger, and the axis limits could be chosen in a way to present the information more clearly."

Following the reviewer's comment, we have enlarged the maps, increased the resolution quality, and modified the axes scales.
The updated Figure 3 (Figure 4 in the revised manuscript) is shown below:

[Figure]

**Figure 4.** Modeled upper bound on carbon export production compared to field observations as a function of mixed layer depth (MLD) and sea surface temperature (SST). (A) The thick gray line represents the upper bound fitted to the net community production (NCP) data. Dash-lines represent the upper bounds calculated using parameters available in the literature (Table 2). (B) NCP as a function of SST with isopleths of constant upper bounds color coded for MLD. NCP observations are color coded with MLD. (C) Surface representing the envelope of the modeled upper bound of carbon export production as a function of SST and MLD. Bars represent field observations color coded with the ratio of NCP to the upper bound. Observations are based on [234]Th and sediment traps estimates of carbon export production and $O_2$/Ar-derived NCP. A stoichiometric ratio of $O_2/C=1.4$ was used to convert NCP from $O_2$ to C units (Laws, 1991). To account for the effect of PAR on export production, both MLD and carbon fluxes are normalized to $-log(1 - I_m(0))$ (see equations (19) and (21)). The temperature dependence of $r_{HR}$ was modeled as $r_{HR} = r_{HR}^0 \times e^{0.08 \times T}$.

Reviewer's comment: "Eq. 21 may be missing a normalizing constant - a proportionality (Eq. 20) is not the same as an equals sign. The values of Pt and Bt both merit a bit more discussion -both numbers have some associated uncertainty, do they not?"

Following the reviewer's comment, we modified equations (20a) and (20b):
$$\mu_{max} = \mu_{max}^0 \times e^{P_t \times T} \qquad (20a)$$
$$r_{HR} = r_{HR}^0 \times e^{B_t \times T} \qquad (20b)$$

We also now elaborate on the uncertainties associated with both parameters. We modified the following paragraph in the section on caveats and limitations:

- $\mu_{max}$ and $r_{HR}$ are influenced by environmental factors other than temperature, including community structure (Chen and Laws, 2017), and may vary with depth within the mixed layer. For these reasons, the equations relating $\mu_{max}$ and $r_{HR}$ (i.e., $B_t$ and $P_t$) to temperature carry uncertainties (Bissinger et al., 2008; Edwards et al., 2016; Kremer et al., 2017; López-Urrutia and Morán, 2007; Rivkin and Legendre, 2001) which impacts our estimates of the upper bound on carbon export, especially in warmer regions. As in other recent studies (Cael and Follows, 2016; Cael et al., 2017; Dutkiewicz et al., 2001; Gong et al., 2015; Gong et al., 2017; Huisman et al., 2006; Taylor and Ferrari, 2011), we model heterotrophic respiration to vary in proportion to phytoplankton concentration. The model could be further improved by explicitly including the concentration of heterotrophs. See point above on the grazing effect on export with regards to $r_{HR}$.

**Reviewer 3:**

We thank the reviewer for his/her insightful review of our manuscript. Below, we provide a response to the reviewer's comments which we think have significantly improved our manuscript.

Reviewer's comment: "The paper would benefit from more motivation for the model at the start. The introduction is fairly short and general. The reader would be more eager to dive into all the details of the model if the need for this model and the questions that the authors hope to address with it were clearly laid out near the beginning of the paper.

Figure 3 demonstrates that there are patterns in the observations that we should seek to explain, but this is only briefly introduced at the start of the paper. Figure 4 shows intuitive results, so here too the motivation to do the global analysis should be specifically stated."

Following the reviewer's comment, we now discuss the relevance of the study at the end of the third paragraph in the introduction:

Likewise, the effects of light and nutrient on carbon fluxes are difficult to disentangle. For example, high-nutrient, low-chlorophyll regimes in the Southern Ocean have been attributed to iron limitation (Boyd et al., 2000), deep mixed layers and light limitation (Nelson and Smith, 1991; Mitchell and Holm-Hanse, 1991; Mitchell et al., 1991), or both (Sunda and Huntsman, 1997). To decompose the influence of light and nutrient availability on NCP, we define the upper bound on carbon export from the mixed layer ($NCP^*$) as the maximum export achievable should all limiting factors other than light (taking into account self-shading) be alleviated.

Reviewer's comment: "A large proportion of export is potentially controlled by bloom dynamics as phytoplankton escape heterotrophic grazing control or not. The proposed model misses these dynamics by forcing heterotrophic respiration to be solely proportional to phytoplankton concentration, rather than also include heterotroph concentrations. Of course, this simplifies the model considerably. However, this simplification may render the results irrelevant since the model then does not approximate the real system closely enough. At the very least, the authors need to carefully argue that their model remains valid for the questions they wish to address despite this simplification of heterotrophic respiration. Such an argument is presently missing from the paper."

We now better acknowledge this limitation in our revised manuscript, including in the section on caveats and limitations where we expand on grazing and heterotrophy. We now also cite additional papers where a similar approach has been used (e.g., Cael and Follows, 2016, Cael et al., 2017, Dutkiewicz et al., 2001, Gong et al., 2015, Gong et al., 2017, Huisman et al., 2006, and Taylor and Ferrari, 2011).

In the section on caveats and limitations, we added the following paragraph:
- $\mu_{max}$ and $r_{HR}$ are influenced by environmental factors other than temperature, including community structure (Chen and Laws, 2017), and may vary with depth within the mixed layer. For these reasons, the equations relating $\mu_{max}$ and $r_{HR}$ (i.e., $B_t$ and $P_t$) to temperature carry uncertainties (Bissinger et al., 2008; Edwards et al., 2016; Kremer et al., 2017; López-Urrutia and Morán, 2007; Rivkin and Legendre, 2001) which impacts our estimates of the upper bound on carbon export, especially in warmer regions. As in other recent studies (Cael and Follows, 2016; Cael et al., 2017; Dutkiewicz et al., 2001; Gong et al., 2015; Gong et al., 2017; Huisman et al., 2006; Taylor and Ferrari, 2011), we model heterotrophic respiration to vary in proportion to phytoplankton concentration. The model could be further improved by explicitly including the concentration of heterotrophs. See point above on the grazing effect on export with regards to $r_{HR}$.

Reviewer's comment: "I would like to see more clarity about how the generalized conclusions of the model depend on choices for specific constants. For example, the discussion in the paragraph beginning on line 121 only holds where kc is significant. As kc goes toward zero, selfshading decreases and NPP will continuously increase as C increases. The text is not clear on whether the kc required to cause the self-shading induced decrease in dNCP/dC above a certain C is reasonable. The paper discusses specific values for some of these constants later in section 2.5, but it seems as though the values of these constants affect earlier conclusions as well."

Because pure water and phytoplankton attenuate light, $K_I^w$ and $k_c$ must be greater than zero. Over the range of $k_c$ values reported in the literature, the behavior of dNCP/dC is not expected to change, as now clarified in the manuscript. Following the reviewer's comment, we now also include a new table (Table 2) which shows the value or range of values (and references) associated with the constants used.

Reviewer's comment: "The simplification in the last part of equation 15 appears to remove the dependence of average mixed layer irradiance on the depth of the mixed layer. Equation 16, based on this simplification, demonstrates that only the respiration term is now sensitive to the mixed layer depth (MLD cancels from the first term). This seems to run counter to all the previous arguments that MLD is important to integrated NPP values."

This is an important point raised by the reviewer. We have now revised the approximation in Equation (15). Below, we show a figure showing the comparison of upper bounds derived using the original and approximated models. As can be seen, the difference in behavior is small. However, we now include this figure in the manuscript.

[Figure]

**Figure 3**. Upper bounds derived using the original and approximated models. The upper bound for the original model (equations (8-10)) is estimated through a non-linear optimization approach. The upper bound for the approximated model is calculated analytically from equation (19). The models use the constants listed in Table 2 and $I_m(0) = 0.9$. Decreasing $I_m(0)$ and increasing $r_{HR}$ results in greater discrepancies between the original and approximated models in regions with shallow mixed layers.

Reviewer's comment: "Lines 51-56: The discussion of attribution of these patterns seems too limited. Low NCP at high temperatures could be primarily a function of a tendency toward increased stratification and nutrient limitation in warm waters. Additionally, deep mixed layers can bias the $O_2/Ar$ method low if entrainment of deeper waters brings low oxygen into the mixed layer."

In the section on caveats and limitations, we mention that the field observations harbor significant uncertainties. In the same bullet point, we now mention as an example that "deep mixed layers can bias the $O_2/Ar$ method low if entrainment of deeper waters brings low $O_2$ into the mixed layer".

On line 275 of the original manuscript, we now further elaborate on the low $f_{pt}$ in warm waters. These waters cannot reach their full export potential because of increased stratification and nutrient limitation ("The ultra-oligotrophic subtropical waters have a low export ratio, a strong biological pump efficiency with exhaustion of nutrients at the ocean surface, and therefore have not reached their full light potential (low $f_{pt}$) because of the strong stratification and nutrient limitation").

Reviewer's comment: "Line 82: "light" attenuation coefficient rather than "diffusion" attenuation coefficient?"

The term "light" attenuation coefficient has been replaced with "diffusion" attenuation coefficient.

Reviewer's comment: "Lines 113-120 and following paragraph: This section is unclear in places. Figure 2 could be actively discussed to demonstrate why dNCP/dC asymptotes at −r*MLD through comparison of the production and respiration terms on the right side of Figure 2a where the production term becomes stable. I spent a long time thinking about this, so the authors could really lead the reader through these arguments better. The text implies in places that dNCP/dC always decreases with increasing C (lines 113-114), but this is only the case at C larger than C*."

As stated in our original manuscript, dNCP/dC systematically decreases with increasing C (this is because $\frac{d^2 NCP(0,MLD)}{dC^2}$ is smaller than zero (see equation 12)). However, dNCP/dC remains positive below C*, and becomes negative above C*. Following the reviewer's comment, we now discuss the asymptote of $\frac{dNCP(0,MLD)}{dC}$ using Figure 2: "Because increasing $C$ decreases light availability due to self-shading, $NPP(0,MLD)$ saturates with increasing $C$. Thus, $NCP(0,MLD)$ will reach an asymptote of $\lim_{C \to \infty}\left(\frac{dNCP(0,MLD)}{dC}\right) = -r_{HR} \times MLD < 0$, because $HR(0,MLD)$ linearly increases with increasing $C$ while $NPP(0,MLD)$ plateaus (Figure 2)."

Reviewer's comment: "Lines 138-140: the statement here that integrated NCP is maximized when the MLD is below the compensation depth seems contrary to the schematic representation of the system in Figure 1a vs. 1b where the integrated NCP is maximized at the compensation depth."

The compensation depth is a function of C. In Figure 1, C is assumed to be constant and MLD is allowed to vary (e.g., synoptic variability in MLD). In this case, depth-integrated NCP will be maximized when MLD deepens or shoals to the compensation depth. Conversely, in (equation 14), C is allowed to vary for a given MLD (e.g., stable water column with varying phytoplankton biomass), in which case, the compensation depth will respond and the depth-integrated NCP peaks when the mixed layer is slightly deeper than the compensation depth.

We amended the manuscript with the following sentence: "We note that in equation (14) the NCP profile ($NCP(z)$) varies with increasing $C$, which is different from what is conceptually presented in Figure 1. The depth-integrated NCP in Figure 1 maximizes at the compensation depth because the NCP profile ($NCP(z)$) is assumed to be invariant."

Reviewer's comment: "Line 163: Why the MLD should satisfy the given conditions are not clear here until Line 171, where the authors state that they have chosen not to consider other possibilities."

Following the reviewer's comment, we reorganized the sentences:
"Equation (18) decreases with MLD. As $C^*$ is positive ($C^* \geq 0$) and cannot go to infinity ($C^* \leq C^*_{max}$), MLD should satisfy $MLD_{C^*_{max}} \leq MLD \leq \frac{\mu^*}{r_{HR} \times K_I^w}$, where $MLD_{C^*_{max}}$ represents the MLD corresponding to the maximum achievable autotroph's biomass concentration ($C^*_{max}$) in the surface ocean. The $NCP^*$ model for $0 \leq MLD < MLD_{C^*_{max}}$ is not discussed here, because we do not have data with very shallow MLD to constrain and evaluate the model. The derivation of the model is however presented in the supplementary material."

Reviewer's comment: "Equations 20a and 20b: These are written as simple proportionalities here, but later treated as though the proportional sign is replaced with an equal sign. It seems like there should be an additional constant."

Following the reviewer's comment, we modified equations (20a) and (20b):
$$\mu_{max} = \mu^0_{max} \times e^{P_t \times T} \qquad (20a)$$
$$r_{HR} = r^0_{HR} \times e^{B_t \times T} \qquad (20b)$$

Reviewer's comment: "Section 2.5: Where specific values or ranges of values are chosen for model constants, it would be helpful to list these in the table defining notation."

Following the reviewer's comment, we added a table that includes the typical range of the parameters with references.

**Table 2**. Value or range of values with references for the parameters used in the model

| Parameter | Range or value | Reference |
|---|---|---|
| $K_I^w$ | 0.09 | (Werdell and Bailey, 2005) |
| $k_c$ | 0.03 | (Werdell and Bailey, 2005) |
| Carbon to chlorophyll ratio | 90 | (Arrigo et al., 2008) |
| $k_m^I$ | 4.1 Einstein m$^{-2}$ d$^{-1}$ | (Behrenfeld and Falkowski, 1997) |
| $P_t$ | 0.0663 | (Eppley, 1972) |
| $B_t$ | 0.08 | (Rivkin and Legendre, 2001; López-Urrutia et al., 2006) |
| $\mu_{max}$ | 1 d$^{-1}$, 1.2 d$^{-1}$ | (Laws et al., 2000; Eppley, 1972) |
| $r_{HR}$ | 0.1 d$^{-1}$, 0.2 d$^{-1}$ | (Laws et al., 2000; Mitchell et al., 1991) |

Reviewer's comment: "Line 196: It's unclear why data could be below the theoretical line due to light limitation, when the theoretical line is specifically modeled to include light limitation."

Following the reviewer's comment, we have removed the reference to light limitation in the sentence. Now: "Conversely, points below the upper bound may be nutrient limited."

Reviewer's comment: "Model-data differences are difficult to clearly discern in Figure 3b. Perhaps it would be useful to directly plot model-data differences in a third panel. That the NCP* model performs poorly in warm deep mixed layers (as stated on lines 210-211) cannot be clearly seen in the figure."

Following the reviewer's comment, we added a panel in the original Figure 3 (now Figure 4) showing the upper bound as a function of SST with isopleths of constant upper bounds color coded for MLD.

[Figure]

**Figure 4.** Modeled upper bound on carbon export production compared to field observations as a function of mixed layer depth (MLD) and sea surface temperature (SST). (A) The thick gray line represents the upper bound fitted to the net community production (NCP) data. Dash-lines represent the upper bounds calculated using parameters available in the literature (Table 2). (B) NCP as a function of SST with isopleths of constant upper bounds color coded for MLD. NCP observations are color coded with MLD. (C) Surface representing the envelope of the modeled upper bound of carbon export production as a function of SST and MLD. Bars represent field observations color coded with the ratio of NCP to the upper bound. Observations are based on [234]Th and sediment traps estimates of carbon export production and $O_2$/Ar-derived NCP. A stoichiometric ratio of $O_2$/C=1.4 was used to convert NCP from $O_2$ to C units (Laws, 1991). To account for the effect of PAR on export production, both MLD and carbon fluxes are normalized to $-log(1 - I_m(0))$ (see equations (19) and (21)). The temperature dependence of $r_{HR}$ was modeled as $r_{HR} = r_{HR}^0 \times e^{0.08 \times T}$.

Reviewer's comment: "Line 281: The text discusses discrepancies between predicted and observed NCP*. However, only NCP can be observed, not NCP*."

[revised manuscript text omitted]

de Boyer Montegut, C., Madec, G., Fischer, A. S., Lazar, A., and Iudicone, D.: Mixed layer depth over the global ocean: An examination of profile data and a profile-based climatology, Journal of Geophysical Research-Oceans, 109, doi:10.1029/2004JC002378, 2004.

Dong, S., J. Sprintall, Gille, S. T., and Talley, L.: Southern Ocean mixed-layer depth from Argo float profiles, Journal of Geophysical Research-Oceans, 113, doi:10.1029/2006JC004051, 2008.

Eveleth, R., Timmermans, M. L., and Cassar, N.: Physical and biological controls on oxygen saturation variability in the upper Arctic Ocean, Journal of Geophysical Research-Oceans, 119, 7420-7432, doi:10.1002/2014JC009816, 2014.

Eveleth, R., Cassar, N., Sherrell, R. M., Ducklow, H., Meredith, M., Venables, H., Lin, Y., and Li, Z.: Ice melt influence on summertime net community production along the Western Antarctic Peninsula, Deep Sea Research Part II., 139, 89-102, doi:10.1016/j.dsr2.2016.07.016, 2017.

Hamme, R. C., Cassar, N., Lance, V. P., Vaillancourt, R. D., Bender, M. L., Strutton, P. G., Moore, T. S., DeGrandpre, M. D., Sabine, C. L., Ho, D. T., and Hargreaves, B. R.: Dissolved $O_2$/Ar and other methods reveal rapid changes in productivity during a Lagrangian experiment in the Southern Ocean, Journal of Geophysical Research-Oceans, 117, doi:10.1029/2011JC007046, 2012.

Huang, K., Ducklow, H., Vernet, M., Cassar, N., and Bender, M. L.: Export production and its regulating factors in the West Antarctica Peninsula region of the Southern Ocean, Global Biogeochem Cy, 26, doi:10.1029/2010GB004028, 2012.

Jonsson, B. F., Doney, S. C., Dunne, J. P., and Bender, M. L.: Evaluation of the Southern Ocean $O_2$/Ar-based NCP estimates in a model framework, Journal of geophysical Research, 118, 385-399, doi:10.1002/jgrg.20032, 2013.

Juranek, L. W., Hamme, R. C., Kaiser, J., Wanninkhof, R., and Quay, P. D.: Evidence of $O_2$ consumption in underway seawater lines: Implications for air-sea $O_2$ and $CO_2$ fluxes, Geophys Res Lett, 37, doi:10.1029/2009GL040423, 2010.

Li, Z. and Cassar, N.: Satellite estimates of net community production based on $O_2$/Ar observations and comparison to other estimates, Global Biogeochem Cy, 30, 735-752, doi:10.1002/2015GB005314, 2016.

Martin, J. H., Knauer, G. A., Karl, D. M., and Broenkow, W. W.: VERTEX: Carbon Cycling in the Northeast Pacific, Deep Sea Research Part A., 34, 267-285, doi:10.1016/0198-0149(87)90086-0, 1987.

Martin, P., van der Loeff, M. R., Carssar, N., Vandromme, P., d'Ovidio, F., Stemmann, L., Rengarajan, R., Soares, M., González, H. E., Ebersbach, F., Lampitt, R. S., Sanders, R., Barnett, B. A., Smetacek, V., and Naqvi, S. W. A.: Iron fertilization enhanced net community production but not downward particle flux during the Southern Ocean iron fertilization experiment LOHAFEX, Global Biogeochem Cy, 27, 871-881, doi:10.1002/gbc.20077, 2013.

Mouw, C. B., Barnett, A., McKinley, G. A., Gloege, L., and Pilcher, D.: Global ocean particulate organic carbon flux merged with satellite parameters, Earth System Science Data, 8, 531-541, doi:10.5194/essd-8-531-2016, 2016.

Reuer, M. K., Barnett, B. A., Bender, M. L., Falkowski, P. G., and Hendricks, M. B.: New estimates of Southern Ocean biological production rates from $O_2$/Ar ratios and the triple isotope composition of $O_2$, Deep Sea Research Part I, 54, 951-974, doi:10.1016/j.dsr.2007.02.007, 2007.

Shadwick, E. H., Tilbrook, B., Cassar, N., Trull, T. W., and Rintoul, S. R.: Summertime physical and biological controls on $O_2$ and $CO_2$ in the Australian Sector of the Southern Ocean, J Marine Syst, 147, 21-28, doi:10.1016/j.jmarsys.2013.12.008, 2015.

Stanley, R. H. R., Kirkpatrick, J. B., Cassar, N., Barnett, B. A., and Bender, M. L.: Net community production and gross primary production rates in the western equatorial Pacific, Global Biogeochem Cy, 24, doi:10.1029/2009GB003651, 2010.

Tortell, P. D., Gueguen, C., Long, M. C., Payne, C. D., Lee, P., and DiTullio, G. R.: Spatial variability and temporal dynamics of surface water $pCO_2$, $\Delta O_2$/Ar and dimethylsulfide in the Ross Sea, Antarctica, Deep Sea Research Part I, 58, 241-259, doi:10.1016/j.dsr.2010.12.006, 2011.

Wanninkhof, R.: Relationship between wind speed and gas exchange over the Ocean, Journal of Geophysical Research-Oceans, 97, 7373-7382, doi:10.1029/92JC00188, 1992.

Werdell, P. J. and Bailey, S. W.: An improved in-situ bio-optical data set for ocean color algorithm development and satellite data product validation, Remote Sensing of Environment, 98, 122-140, doi:10.1016/j.rse.2005.07.001, 2005.

---

## Author Response (AR2)

Thursday, September 28, 2017

Jack Middelburg
Editor
Biogeosciences

Dear Dr. Middelburg,

We would first like to thank you for your careful examination of our manuscript. We have taken into account your comments in the revised manuscript.

Below is a response to your comments and the revised manuscript. Please do not hesitate to contact us should you have any additional questions or comments on our manuscript.

Sincerely,
Zuchuan Li

Division of Earth and Ocean Sciences
Nicholas School of the Environment
Duke University
email. zuchuan.li@duke.edu

[Figure]

We thank the editor for his careful review of our manuscript. Below, we provide a response to the editor's comments.

line 32: meta-analysis (for readability).
Following the editor's comment, "meta analysis" has been replaced with "meta-analysis".

line 65, 344: replace on the other hand with however or alike because there is no on the one hand (and they always go together).
Following the editor's comment, "on the other hand" has been replaced with "however".

line 112: are assumed to be constant/uniform. The logic of calling half-saturation constants well mixed is not clear. Constant is also what matters.
Following the editor's comment, "well mixed" has been replaced with "constant/uniform".

[revised manuscript text omitted]

de Boyer Montegut, C., Madec, G., Fischer, A. S., Lazar, A., and Iudicone, D.: Mixed layer depth over the global ocean: An examination of profile data and a profile-based climatology, Journal of Geophysical Research-Oceans, 109, doi:10.1029/2004JC002378, 2004.

Dong, S., J. Sprintall, Gille, S. T., and Talley, L.: Southern Ocean mixed-layer depth from Argo float profiles, Journal of Geophysical Research-Oceans, 113, doi:10.1029/2006JC004051, 2008.

Eveleth, R., Timmermans, M. L., and Cassar, N.: Physical and biological controls on oxygen saturation variability in the upper Arctic Ocean, Journal of Geophysical Research-Oceans, 119, 7420-7432, doi:10.1002/2014JC009816, 2014.

Eveleth, R., Cassar, N., Sherrell, R. M., Ducklow, H., Meredith, M., Venables, H., Lin, Y., and Li, Z.: Ice melt influence on summertime net community production along the Western Antarctic Peninsula, Deep Sea Research Part II., 139, 89-102, doi:10.1016/j.dsr2.2016.07.016, 2017.

Hamme, R. C., Cassar, N., Lance, V. P., Vaillancourt, R. D., Bender, M. L., Strutton, P. G., Moore, T. S., DeGrandpre, M. D., Sabine, C. L., Ho, D. T., and Hargreaves, B. R.: Dissolved $O_2$/Ar and other methods reveal rapid changes in productivity during a Lagrangian experiment in the Southern Ocean, Journal of Geophysical Research-Oceans, 117, doi:10.1029/2011JC007046, 2012.

Huang, K., Ducklow, H., Vernet, M., Cassar, N., and Bender, M. L.: Export production and its regulating factors in the West Antarctica Peninsula region of the Southern Ocean, Global Biogeochem Cy, 26, doi:10.1029/2010GB004028, 2012.

Jonsson, B. F., Doney, S. C., Dunne, J. P., and Bender, M. L.: Evaluation of the Southern Ocean $O_2$/Ar-based NCP estimates in a model framework, Journal of geophysical Research, 118, 385-399, doi:10.1002/jgrg.20032, 2013.

Juranek, L. W., Hamme, R. C., Kaiser, J., Wanninkhof, R., and Quay, P. D.: Evidence of $O_2$ consumption in underway seawater lines: Implications for air-sea $O_2$ and $CO_2$ fluxes, Geophys Res Lett, 37, doi:10.1029/2009GL040423, 2010.

Li, Z. and Cassar, N.: Satellite estimates of net community production based on $O_2$/Ar observations and comparison to other estimates, Global Biogeochem Cy, 30, 735-752, doi:10.1002/2015GB005314, 2016.

Martin, J. H., Knauer, G. A., Karl, D. M., and Broenkow, W. W.: VERTEX: Carbon Cycling in the Northeast Pacific, Deep Sea Research Part A., 34, 267-285, doi:10.1016/0198-0149(87)90086-0, 1987.

Martin, P., van der Loeff, M. R., Carssar, N., Vandromme, P., d'Ovidio, F., Stemmann, L., Rengarajan, R., Soares, M., González, H. E., Ebersbach, F., Lampitt, R. S., Sanders, R., Barnett, B. A., Smetacek, V., and Naqvi, S. W. A.: Iron fertilization enhanced net community production but not downward particle flux during the Southern Ocean iron fertilization experiment LOHAFEX, Global Biogeochem Cy, 27, 871-881, doi:10.1002/gbc.20077, 2013.

Mouw, C. B., Barnett, A., McKinley, G. A., Gloege, L., and Pilcher, D.: Global ocean particulate organic carbon flux merged with satellite parameters, Earth System Science Data, 8, 531-541, doi:10.5194/essd-8-531-2016, 2016.

Reuer, M. K., Barnett, B. A., Bender, M. L., Falkowski, P. G., and Hendricks, M. B.: New estimates of Southern Ocean biological production rates from $O_2$/Ar ratios and the triple isotope composition of $O_2$, Deep Sea Research Part I, 54, 951-974, doi:10.1016/j.dsr.2007.02.007, 2007.

Shadwick, E. H., Tilbrook, B., Cassar, N., Trull, T. W., and Rintoul, S. R.: Summertime physical and biological controls on $O_2$ and $CO_2$ in the Australian Sector of the Southern Ocean, J Marine Syst, 147, 21-28, doi:10.1016/j.jmarsys.2013.12.008, 2015.

Stanley, R. H. R., Kirkpatrick, J. B., Cassar, N., Barnett, B. A., and Bender, M. L.: Net community production and gross primary production rates in the western equatorial Pacific, Global Biogeochem Cy, 24, doi:10.1029/2009GB003651, 2010.

Tortell, P. D., Gueguen, C., Long, M. C., Payne, C. D., Lee, P., and DiTullio, G. R.: Spatial variability and temporal dynamics of surface water $pCO_2$, $\Delta O_2$/Ar and dimethylsulfide in the Ross Sea, Antarctica, Deep Sea Research Part I, 58, 241-259, doi:10.1016/j.dsr.2010.12.006, 2011.

Wanninkhof, R.: Relationship between wind speed and gas exchange over the Ocean, Journal of Geophysical Research-Oceans, 97, 7373-7382, doi:10.1029/92JC00188, 1992.

Werdell, P. J. and Bailey, S. W.: An improved in-situ bio-optical data set for ocean color algorithm development and satellite data product validation, Remote Sensing of Environment, 98, 122-140, doi:10.1016/j.rse.2005.07.001, 2005.